# BiRNA-BERT allows efficient RNA language modeling with adaptive tokenization

Md Toki Tahmid[1,4], Haz Sameen Shahgir[2,4], Sazan Mahbub[3], Yue Dong[2] & Md Shamsuzzoha Bayzid [1] ✉

Transformer-based models have achieved remarkable success in biological sequence modeling, yet their application to RNA remains constrained by sequence length limitations. Existing RNA language models often truncate inputs, discarding distal nucleotide context crucial for full-length tasks. Additionally, advanced NLP tokenization methods do not directly apply to biological sequences, where nucleotide-level resolution is essential for tasks like secondary structure prediction. To address these challenges, we introduce BiRNA-BERT, a 117M-parameter Transformer encoder trained on 36 million non-coding RNA sequences. At its core is an adaptive dual-tokenization framework that combines nucleotide-level (NUC) encoding for fine-grained structural tasks with byte-pair encoding (BPE) for efficient long-sequence processing. BiRNA-BERT dynamically selects tokenization based on input length, enabling it to process arbitrarily long sequences without truncation. We demonstrate state-of-the-art performance across tasks ranging from short-sequence classification to long-context modeling and fine-grained nucleotide level RNA structural prediction. Our information-theoretic analysis reveals the trade-offs between BPE compression and NUC tokenization, which we again validate empirically. Finally, BiRNA-BERT achieves strong intrinsic language modeling performance–measured by perplexity and token recovery–while remaining more compact than existing RNA models. The code and model weights are available at https://github.com/buetnlpbio/ BiRNA-BERT.

The introduction of encoder-only transformer models has revolutionized Natural Language Processing (NLP) by significantly improving our ability to extract deep semantic representations from text, which can be used for a wide range of downstream tasks[1,2]. This success has inspired researchers to apply the pretraining-finetuning paradigm to a wide range of topics beyond NLP, including biological sequence modeling with remarkable success[3-7]. However, transferring improvements from natural language processing (NLP) to biological sequences is not always straightforward. A key challenge arises from the need to model extremely long sequences for certain tasks, as biological sequences can range from tens to millions of nucleotides or amino acids. The standard transformer encoder architecture (e.g., BERT[1]) uses positional embeddings that limit the maximum input length to usually 512 or 1024, requiring truncation or inefficient workarounds for biological sequences.

In addition, despite recent advancements in methodological and architectural research efforts to scale up the length for BERT[8-10] by addressing the limited context window[11,12] or with more efficient

tokenization methods[13,14], they have not yet been successfully adopted by bioinformatics without sacrificing performance or sample efficiency. The limited number of approaches to adopting new positional encoding and improved pretraining developed in NLP often fall short in biological sequencing. For example, Nucleotide Transformers (NT)[15] replace nucleotide-level tokenization (NUC) with non-overlapping $k$-mers, where $k$-mers represent subtokens of fixed length $k$, enabling the model to handle sequences that are $k$ times longer while sacrificing granularity. This limitation is acknowledged in DNABERT-2[4], where they demonstrate the poor sample efficiency of $k$-mer tokenization and propose adapting byte pair encoding (BPE) tokenization for DNA sequences. However, their approaches still suffer from granularity loss by combining common subsequences into one token, and the original paper does not provide a controlled comparison between NUC and BPE. Moreover, in domains like RNA or protein sequences where secondary structure prediction or granular alignment tasks are important, simply using BPE tokenization would prevent the trained

[1]Department of Computer Science and Engineering, Bangladesh University of Engineering and Technology, Dhaka, Bangladesh. [2]Department of Computer Science and Engineering, University of California Riverside, Riverside, CA, USA. [3]Department of Computer Science, University of Maryland at College Park, College Park, College Park, MD, USA. [4]These authors contributed equally: Md Toki Tahmid, Haz Sameen Shahgir. ✉e-mail: shams_bayzid@cse.buet.ac.bd

foundation model from generating nucleotide- or amino acid-level predictions, which are essential for these additional tasks.

Following the success of *foundation models*—models pre-trained on vast amounts of unlabeled data—in protein sequences, building similar pre-trained models for DNA/RNA sequences has gained significant attention from bioinformatics researchers. Notable foundation models for DNA include DNABERT[3], DNABERT-2[4], and Nucleotide Transformer[15], all of which are BERT-style encoder-only models largely following the pretrain-then-finetune paradigm. Considerable progress has been made for RNA sequences as well.[5] proposed RNA-FM, a 100 million parameters model pre-trained on 23.7 million unannotated ncRNAs from the RNAcentral database[16] using the standard masked language modeling task and nucleotide-level tokenization (NUC). Uni-RNA[17] was pre-trained on a dataset of 1 billion RNA sequences for a spectrum of structural and functional downstream tasks.[18] proposed a Multiple sequence alignment (MSA)-based RNA language model (RNA-MSM), a BERT-style language model that utilizes a set of homologous sequences per forward propagation and produces attention maps and embeddings that have direct correlations to RNA secondary structure and solvent accessibility without supervised training. For downstream tasks, RNA-MSM finds homologous sequences of the input sequence from a stored database, computes the MSA matrix using RNAcmap[19], and uses information from the MSA as input to the model. However, obtaining the MSAs is a time-consuming procedure, for example, it takes RNAcmap on average 9 hours to obtain an MSA for one RNA sequence of length 60.[6] introduced RiNALMo, a 650M parameter model that incorporates algorithm and architecture improvements in NLP models, namely Rotary Position Encoding (RoPE)[12], FlashAttention2[20], and SwiGLU[21], and is pre-trained on 36 million ncRNA sequences from RNAcentral[16], NT, rfam[22], and Ensembl databases[23]. RiNALMo achieves state-of-the-art performance on a variety of downstream tasks such as splice-site and mean ribosome loading and exhibits stronger generalization than previous thermodynamics-based and deep learning models on secondary structure prediction.

All the aforementioned biological foundation models either use character-level (nucleotide-level) tokenization or BPE tokenization, each with its trade-offs. Approaches using nucleotide-level tokenization inevitably face long tokenized sequences, while approaches using BPE, such as DNABERT-2, do not cater to tasks requiring nucleotide-level granularity. Given these limitations, we propose **Bi**-tokenization **RNABERT** (BiRNA-BERT)—a Transformer encoder model for RNA sequences pretrained on both NUC and BPE tokens of the same RNA sequences simultaneously. BiRNA-BERT uses Attention with Linear Biases (ALiBi)[11] which allows the context window to be extended without retraining and can dynamically choose between NUC and BPE tokenization based on the input sequence length. For shorter sequences, it utilizes NUC to capture fine-grained patterns, and for longer sequences, it switches to more efficient BPE tokenization to reduce memory requirements without truncating the input. This dynamic tokenization and context length expansion allows BiRNA-BERT to enable downstream tasks with arbitrarily long sequences and set state-of-the-art results in long sequence tasks such as miRNA-lncRNA interaction prediction, short sequence tasks such as RNA-protein interaction, and different nucleotide level structural prediction tasks, including secondary structure prediction, 3D distance map prediction, and 3D torsion angle prediction.

We conduct further downstream ablation experiments on RNA data to demonstrate that NUC tokens yield better downstream task performance than BPE when enough resources are available to handle the short sequences. In summary, BiRNA-BERT demonstrates the effectiveness and importance of utilizing proper tokenization tailored to the unique demands of bioinformatics tasks. Pre-training models leveraging our proposed tokenization can achieve significant performance gains across a range of biological sequence analysis tasks. Our main contributions can therefore be summarized as follows:

1. We present dual tokenization pretraining—an effective approach that extends the effective context window of biological foundation models with efficient tokenization while retaining the ability to generate character-level embeddings.
2. Using dual tokenization and ALiBi, we train BiRNA-BERT which achieves absolute state-of-the-art results on downstream tasks that span both long and short sequences compared to 6 × larger models while being trained with 15 × less pretraining compute. BiRNA-BERT can dynamically adjust the tokenization algorithm based on the sequence length and available compute.
3. We provide a rigorous mathematical analysis from an information-theoretic perspective about the absolute information loss of token compression with byte pair encoding.
4. We show that NUC outperforms BPE on tasks where sequences are short enough to fit into GPU memory and the trade-off between sequence truncation and information compression works in favor of the latter when long sequences are considered.

## Results

We developed BiRNA-BERT, a transformer-based RNA language model designed to address key challenges in RNA sequence modeling. The architecture integrates ALiBi relative positional encoding to maintain attention across long sequences, and an adaptive tokenization strategy that applies Byte-Pair Encoding (BPE) for long sequences and nucleotide-level (NUC) tokenization for short sequences. This is combined with a dual pretraining approach to support both sequence-level and nucleotide-level downstream tasks within a single model, improving memory efficiency. Pretraining was performed on 26.42 billion nucleotides from RNAcentral, using the MosaicML framework on eight NVIDIA RTX 4090 GPUs over two epochs. We evaluated BiRNA-BERT across multiple downstream tasks, as shown in Fig. 1, including miRNA-lncRNA interaction prediction, RNA-protein interaction prediction, N6-methyladenosine (m6A) site prediction, and nucleotide-level tasks such as secondary structure, 3D distance map, and torsion angle prediction.

This section presents the results from pretraining BiRNA-BERT and its performance on various unsupervised and supervised downstream tasks.

We performed a number of downstream tasks on varying sequence lengths, as shown in Supplementary Fig. S2. In Section "Unsupervised clustering performance", we discuss the unsupervised clustering performance on BiRNA-BERT on RNA structural family and rfam classification. In Section "Long-sequence task with dynamic tokenization: miRNA-lncRNA interaction prediction", we demonstrate that BiRNA-BERT sets new state-of-the-art results on long sequence RNA-RNA interactions by leveraging dynamic tokenization, which previous language models handled with truncation. In the following sections, we discuss the impact of truncation at various lengths.

We cover short sequence downstream tasks in the section "Short sequence tasks" to establish that dual tokenization training has no drawbacks over conventional training. BiRNA-BERT even significantly surpasses similar-sized models such as RNA-FM and BERTRBP on short-sequence tasks. In the section "Nucleotide-level task", we demonstrate BiRNA-BERT's performance on three downstream tasks: RNA secondary structure prediction, 3D distance map prediction, and 3D torsion angle prediction.

In the section "Information theoretic analysis: BPE tokenization comes with absolute information loss", we provide a rigorous mathematical analysis of the dual tokenization scheme from an information-theoretic perspective, which demonstrates the information loss from the BPE compression and how it compensates for the available GPU memory constraints. Moreover, we perform two different downstream tasks of RNA-Protein interaction and RNA N6-methyladenosine sites prediction to empirically show the validity of the information-theoretic analysis. The section "NUC tokenization is preferable over BPE if memory allows" shows the superior performance of NUC tokenization when the sequence lengths are small and can be fully allocated within the available memory. In addition, we discuss the perplexity and recovery accuracy of different RNA language models in the section "Empirical perplexity analysis of different RNA language models".

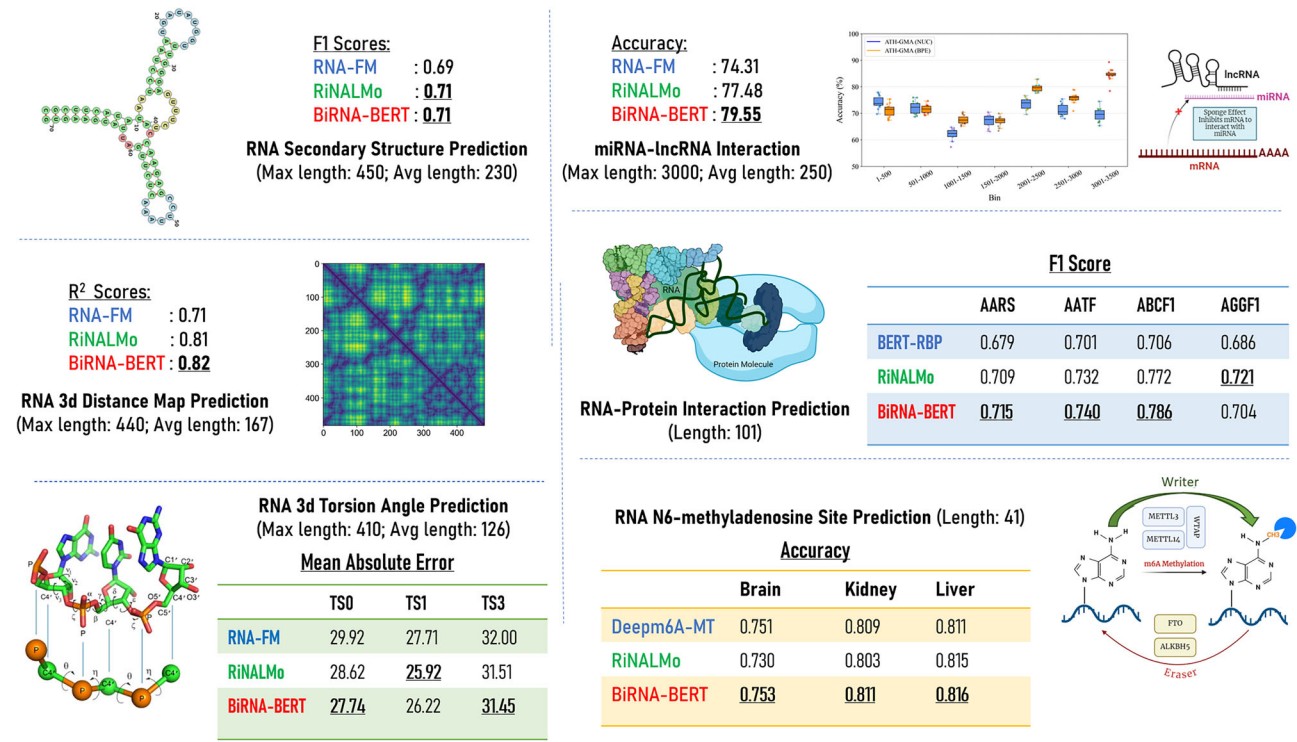

**Fig. 1 | Downstream tasks for assessing the performance of various language models.** The authors acknowledge the use of BioRender.com and Python libraries (Matplotlib and NumPy) for figure preparation.

## Unsupervised clustering performance

To understand the capacity of BiRNA-BERT for unsupervised representation of the embedding space for RNA sequences from different structural families, we performed two different unsupervised clustering tasks. For the first task, we perform unsupervised clustering on 9 RNA structural families: 16s, 23s, 5s, RNaseP, grp1, srp, tRNA, telomerase, tmRNA. We followed the approach adopted by[24], who compiled a dataset of 3,865 RNA sequences from these nine families and used a leave-one-family-out evaluation strategy. RiNALMo similarly adopted this methodology for its clustering and secondary structure performance assessments. To maintain consistency and comparability with prior work, we chose to follow the same strategy.

For the second type of clustering, we consider the rfam dataset[22]. In particular, the rfam version from September 2025 contains 4175 RNA families. Of them, we first cluster the most frequent 30 and 100 classes. Lastly, we consider longest 5000 sequences in the whole dataset which spans 131 classes.

For each RNA language model, after extracting the embeddings, we apply t-SNE for dimensionality reduction and visualize the 2D embedding space. To quantitatively assess the clustering performance of BiRNA-BERT in comparison to other methods on these tasks, we employ the silhouette coefficient as a measure of clustering quality. Silhouette coefficient is a metric used to evaluate the quality of clustering. It measures how similar a data point is to its own cluster compared to other clusters. The coefficient for a data point $i$ is defined as

$$s(i) = \frac{b(i) - a(i)}{\max(a(i), b(i))} \quad (1)$$

Here, $a(i)$ represents the average distance between the data point $i$ and all other points within the same cluster. In contrast, $b(i)$ is the average distance between the data point $i$ and all points in the nearest cluster to which $i$ does not belong. The value of $s(i)$ ranges from $-1$ to 1. A value of 1 indicates that the data point is well-matched to its own cluster and poorly matched to neighboring clusters. A value of 0 indicates that the data point is on or very close to the decision boundary between two neighboring clusters.

Negative values suggest that the data point might have been assigned to the wrong cluster. A higher silhouette coefficient indicates well-defined clusters.

As illustrated in Fig. 2, BiRNA-BERT demonstrates superior clustering performance in RNA structural family classification, achieving a 4.6% improvement over both RNA-FM and RiNALMo. This suggests that the token representations learned by BiRNA-BERT are better aligned with biologically meaningful RNA structure-based groupings.

In contrast, for clustering tasks based on Rfam family annotations, RiNALMo achieves higher performance than BiRNA-BERT in both the 30-class and 100-class settings, with respective gains of 7% and 0.4%. This performance gap can be attributed to differences in the pretraining datasets. RiNALMo was explicitly trained on both the RNACentral and Rfam databases, directly optimizing for features found in Rfam annotations[6]. While BiRNA-BERT was trained solely on the RNACentral corpus–which integrates but does not explicitly emphasize Rfam labels–this discrepancy in dataset composition likely accounts for the observed performance difference. It is worth noting that Rfam models are primarily used in RNACentral for quality control and annotation, not for targeted sequence modeling.

Despite this, the performance difference between BiRNA-BERT and RiNALMo diminishes at larger class counts, suggesting BiRNA-BERT maintains strong generalization capabilities. Furthermore, as discussed in the section "Empirical perplexity analysis of different RNA language models", both models exhibit comparable perplexity and token recovery accuracy, indicating that BiRNA-BERT's representational capacity is not inherently weaker. Rather, its slightly lower performance in Rfam-based clustering is a function of the pretraining objective and data alignment.

Importantly, BiRNA-BERT achieves the highest clustering quality on the subset of the 5000 longest RNA sequences, as shown in the bottom row of Fig. 2. This advantage stems from BiRNA-BERT's ability to handle long sequences without truncation, due to its BPE-based tokenization and extrapolatable positional encoding. In contrast, both RNA-FM and RiNALMo rely on sequence truncation for inputs exceeding their positional limits, thereby losing potentially crucial downstream information.

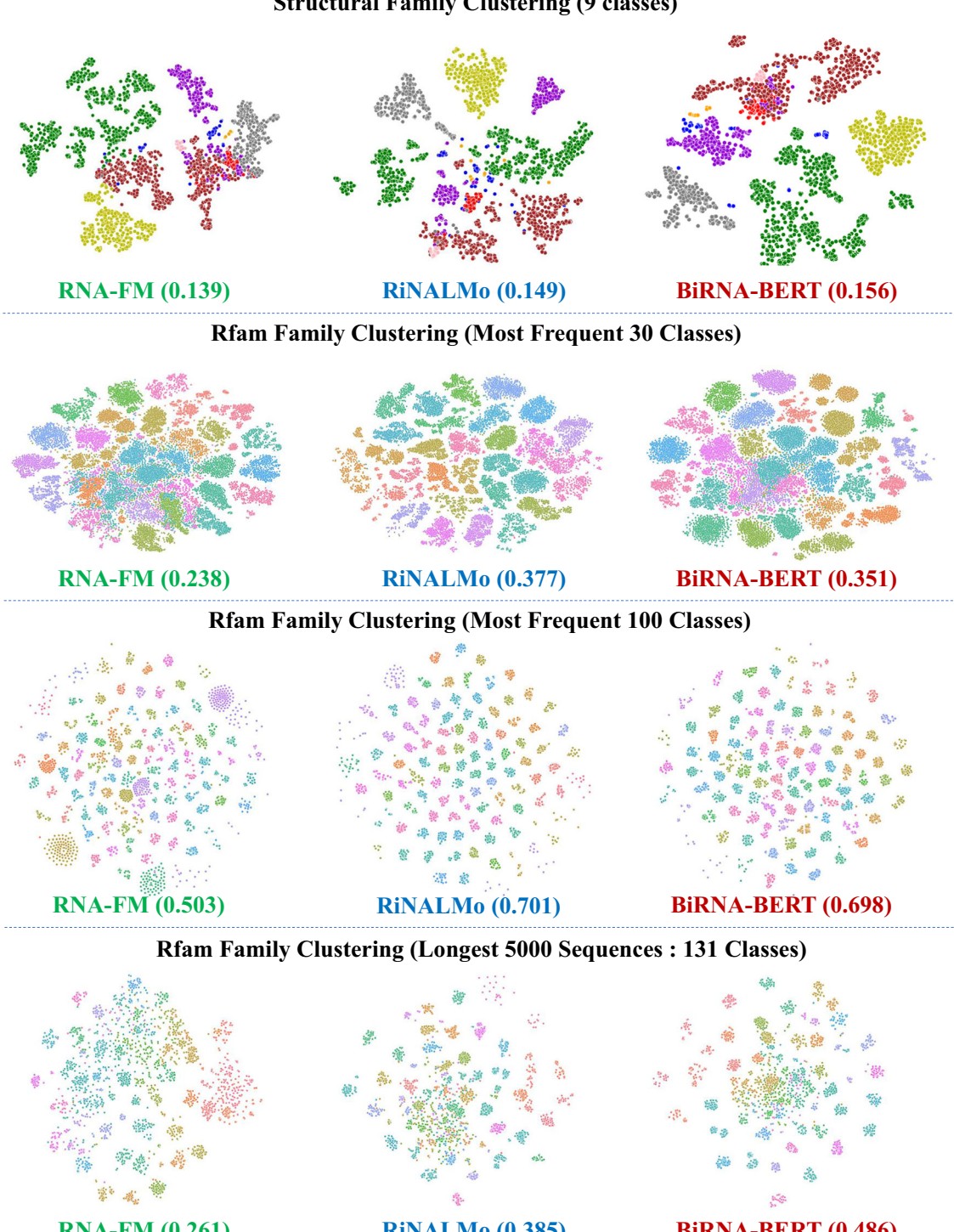

**Fig. 2 | Visualization of unsupervised clustering performance across different RNA language models—RNA-FM, RiNALMo, and BiRNA-BERT—on structural and family-level RNA classification tasks.** The first row presents clustering results based on RNA structural families (9 classes), where BiRNA-BERT achieves the highest score (0.156), indicating more coherent structural grouping. The second and third rows illustrate clustering based on Rfam family labels for the 30 and 100 most frequent classes, respectively, where RiNALMo performs best due to its explicit pretraining on the Rfam database. The final row compares clustering on the longest 5000 sequences (131 Rfam classes), demonstrating BiRNA-BERT's superior performance (0.486), enabled by its long-sequence modeling capability without truncation.

## Long-sequence task with dynamic tokenization: miRNA-lncRNA interaction prediction

We evaluate the interaction between lncRNA and micro RNA (miRNA) to verify the impact of sequence truncation during feature embedding by previous models and compare it with our approach, which avoids sequence cropping to fit computational memory. Instead, we dynamically compress sequence information using our adaptive tokenization scheme.

This is a binary classification task to determine whether a miRNA-lncRNA pair interacts or not. For fine-tuning, we follow the methodology of CORAIN[25] by freezing the backbone encoders, concatenating output

**Table 1 | Accuracy of different models on miRNA-lncRNA dataset**

| Model | Train-Test Dataset | | | | | |
|---|---|---|---|---|---|---|
| | ATH-GMA | ATH-MTR | GMA-ATH | GMA-MTR | MTR-ATH | MTR-GMA |
| CORAIN | 69 | 74 | 67 | 87 | 53 | 71 |
| Pmlipred | 69 | 72 | 69 | 84 | 67 | 85 |
| BioLLMNet | 69 | 74 | 67 | 93 | 58 | 84 |
| RNA-FM | 68.56 | 71.68 | 69.90 | 94.68 | 57.15 | 83.92 |
| RiNALMo | 72.14 | 75.38 | 70.93 | **95.08** | 65.55 | 85.82 |
| BiRNA-BERT (NUC-Truncated) | 71.52 | 76.38 | 70.61 | 94.48 | 62.23 | 83.70 |
| BiRNA-BERT (Adaptive) | **73.02** | **79.42** | **71.97** | 94.00 | **67.83** | **91.06** |

CORAIN[25] is a task-specific CNN-autoencoder and the current state-of-the-art. We fine-tuned RNA-FM, RiNALMo, and both variants of BiRNA-BERT with optimal hyperparameters found using grid search.

features of miRNA and lncRNA, and using a simple convolutional prediction head. We test two strategies using BiRNA: BPE and NUC with truncation, named BiRNA-BPE and BiRNA-NUC. Both strategies use the same BiRNA models and only the input tokenization scheme differs. We always encode miRNA using NUC due to their short lengths. We use NUC with truncation for lncRNA for RNA-FM, RiNALMo, and BiRNA-NUC. We truncate all inputs to 1022 tokens due to the context window limitation of RNA-FM and RiNALMo even though BiRNA-NUC can process longer sequences due to ALiBi. We do not truncate input sequences to BiRNA-BPE since after BPE tokenization the maximum sequence length is 807, still lower than NUC.

The results in Table 1 offer several interesting insights:

1. RNA-FM performs significantly worse than the non-LM-based approach in 4 of 6 test datasets. However, RiNALMo outperforms the current state-of-the-art in all datasets despite sequence truncation. This is noteworthy, as RiNALMo is a six times larger language model than RNA-FM, significantly enhancing its expressive capability.
2. BiRNA-NUC outperforms RNA-FM in 5 out of 6 datasets and provides comparable performance to RiNALMo, despite being the same size as RNA-FM. BiRNA-BPE outperforms RiNALMo by a substantial margin, with improvements of 1.22%, 5.36%, 1.47%, 19.95%, and 6.11%, on the ATH-GMA, ATH-MTR, GMA-ATH, MTR-ATH, and MTR-GMA datasets. It also offers comparable performance in the GMA-MTR dataset, within 1.14% margins. Statistical significance test between RiNALMo and BiRNA-BERT shows that the $p$ value is 0.0004, which verifies the significance of the results obtained.
3. An intuitive way to compare NUC and BPE tokenization is by considering information loss. NUC explicitly truncates sequences to 1022 nucleotides, losing all subsequent information. BPE, on the other hand, compresses the entire sequence (see the section "Information theoretic analysis: BPE tokenization comes with absolute information loss"). In miRNA-lncRNA interaction tasks, we demonstrate that compression (BPE) is preferable to explicit information loss (NUC), while also being more computationally efficient (807 tokens versus 1024 tokens). It is important to note that, in Table 1, BiRNA-BERT (Adaptive) refers to the adaptive tokenization scheme. That is, for short sequences we use NUC tokenization and for long sequences we shift to BPE level tokenization.

BiRNA-BERT uses the information compression technique using BPE and encodes the long sequences with minimal information loss, thereby providing significant performance gain over the traditional models.

## Impact of dual tokenization is more significant in long sequences

To assess the role of tokenization in RNA sequence modeling, we conduct a controlled comparison between nucleotide-level tokenization (NUC) and byte-pair encoding (BPE) across variable sequence lengths. Particularly, we focus on the lncRNA-miRNA interaction prediction task that we discussed before, now focusing on the performance comparison across different length bins. We group the sequences into several bins, with Bin $x - y$ containing sequences with lengths within the range $x$ to $y$. Results are summarized in Fig. 3.

When sequences are truncated to a maximum length of 1024 tokens, as shown in Fig. 3b, the NUC-based model maintains high performance in shorter length bins but exhibits a gradual degradation for longer inputs. This performance decline is attributable to the loss of distal sequence context.

In contrast, Fig. 3c illustrates that BPE tokenization substantially alleviates this issue. By encoding recurring motifs and subsequences as single tokens, BPE effectively reduces the sequence length while preserving semantic integrity. This enables the model to retain information from the full-length input, particularly benefiting long sequences where truncation would otherwise be necessary under an NUC-based scheme.

Figure 3d quantifies the absolute performance difference (BPE minus NUC) across length bins. Notably, in the ATH-GMA dataset (Fig. 3), the BPE-based model outperforms its NUC counterpart by more than 15% in the 3001–3500 bin, where NUC suffers the most from truncation-induced information loss. As shown in Fig. 3, while the performance difference remains marginal or even negative in shorter bins (e.g., 1–500), it becomes increasingly positive for longer sequences.

## BiRNA-BERT significantly outperforms in extremely long sequence task

To rigorously evaluate RNA language models under extremely long sequence settings, we constructed a novel benchmark classification task based on the RNAcentral non-coding RNA database. This dataset comprises sequences with an average length of approximately 2500 nucleotides, ranging up to 10,000 nucleotides, and follows an exponential distribution of lengths. Each sequence is labeled with one of seven species: *Bos taurus*, *Gallus gallus*, *Gorilla gorilla*, *Homo sapiens*, *Mus musculus*, *Pan troglodytes*, and *Rattus norvegicus*. To the best of our knowledge, this dataset has not been included in any prior studies, and we have made it publicly available to facilitate future research on long-sequence RNA modeling.

The task involves classifying these long RNA sequences into their respective species, posing a significant challenge due to the extensive length and complexity of the inputs. Existing RNA language models, such as RiNALMo and RNA-FM, are constrained by their maximum sequence length limits (typically 1024 tokens), necessitating truncation that leads to substantial loss of information. As a result, these models show a notable deterioration in classification performance, achieving F1-scores of 0.620 and 0.532, respectively, as shown in Table 2.

In contrast, BiRNA-BERT demonstrates a substantial improvement, achieving an F1-score of 0.804 on this benchmark. This performance highlights BiRNA-BERT's ability to effectively model extremely long sequences without truncation, thanks to its adaptive tokenization scheme and architectural optimizations tailored for long-context understanding.

These results establish BiRNA-BERT as a more reliable RNA foundation model under high-length constraints and suggest its potential applicability to real-world genomic tasks where full-length sequences must be retained for meaningful interpretation.

## Short sequence tasks

In addition to handling long sequence tasks with an efficient tokenization scheme and relative positional encoding, BiRNA-BERT also demonstrates superior performance in short sequence tasks. Short sequence tasks deal with sequence-level prediction tasks where the sequence length is shorter so that truncation is not necessary. To demonstrate the performance of BiRNA-BERT in short sequence tasks, we finetune BiRNA-BERT for RNA-Protein interaction and RNA N6-methyladenosine site prediction.

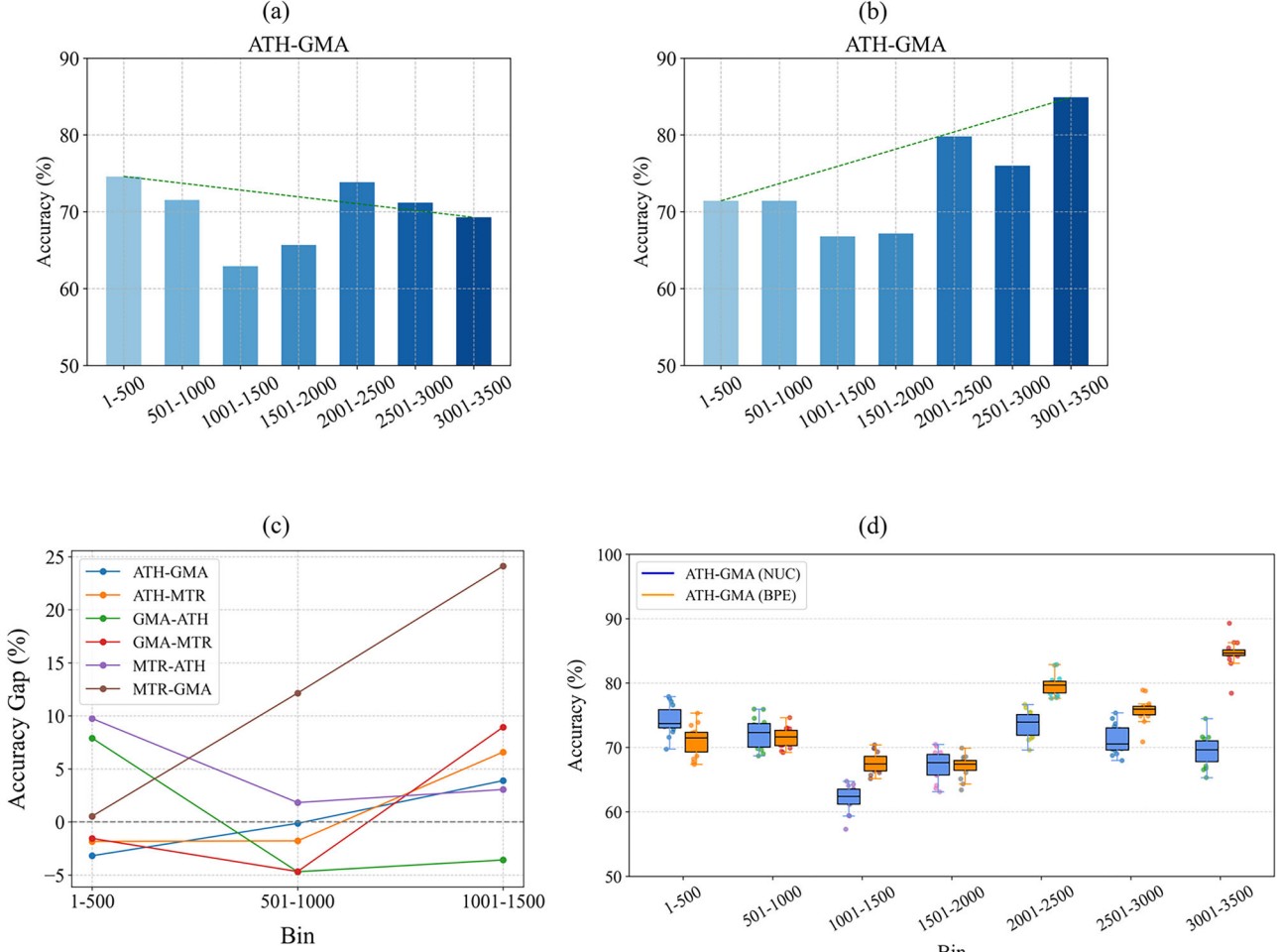

**Fig. 3 | Impact of input sequence length on model performance using NUC and BPE tokenization strategies. a** When sequences are truncated to a fixed length of 1024 during training, performance degradation is observed for longer sequences due to information loss. **b** Using BPE tokenization allows representation of longer sequences without truncation, mitigating performance drops observed in (**a**). **c** On the ATH-GMA dataset, the performance gap widens significantly with increasing sequence length, with BPE outperforming NUC by over 15% in the longest bin.

**d** Comparison of model accuracy between BPE and NUC tokenization across two adjacent sequence length bins: 501–1000 (non-truncated) and 1001–1500 (truncated for NUC). This zoomed-in analysis highlights the immediate impact of truncation on NUC-based performance. While NUC performs well in the non-truncated bin, its accuracy drops significantly in the next bin due to information loss. BPE, unaffected by truncation, maintains higher accuracy in longer sequences.

**Table 2 | F1-scores of RNA language models on the extremely long sequence species classification task**

| Model | F1 Score |
|---|---|
| BiRNA-BERT | 0.804 |
| RiNALMo | 0.620 |
| RNA-FM | 0.532 |

**RNA protein interaction**. RNA-protein interactions are central to understanding post-transcriptional gene regulation, as RNA-binding proteins (RBPs) modulate splicing, transport, degradation, and translation. Predicting whether an RNA sequence binds to a given RBP is a biologically meaningful classification task. Since the input sequences in this task are of fixed length (101 nucleotides), it is naturally formulated as a short sequence prediction problem.

For this study, we focus on five diverse RBPs–AARS, AATF, ABCF1, AGGF1, and AKAP1—each with high-quality experimental annotations. Table 3 summarizes the performance of four transformer-based models on these datasets: BiRNA-BERT (our model), RiNALMo, BERT-RBP, and RNA-BERT. Our model, BiRNA-BERT, consistently outperforms the baselines across nearly all datasets in terms of F1 score, a metric that balances

precision and recall and is especially crucial for imbalanced biological data. Specifically, BiRNA-BERT achieves the best F1 scores for 4 out of 5 proteins, with margins of improvement ranging from +1.1% to +7.6% over the next-best model. For instance, On ABCF1, BiRNA-BERT achieves an F1 of 0.7860, improving upon RiNALMo by +1.8% and outperforming BERT-RBP by +7.9%.

On AARS, it achieves 0.7157, surpassing RiNALMo by +0.9% and BERT-RBP by +5.3%. Even in the most competitive cases such as AKAP1, BiRNA-BERT maintains a strong F1 of 0.7224, just behind RiNALMo, yet significantly better than BERT-RBP and RNA-BERT (both below 0.68). Statistical significance test confirms that the $p$ value between BiRNA-BERT and RiNALMo is 0.0024 in terms of F1-score. While RiNALMo excels in recall, indicating stronger sensitivity in identifying true binding sequences, BiRNA-BERT demonstrates superior overall balance.

**RNA N6-methyladenosine site prediction**. RNA N6-methyladenosine site prediction focuses on predicting N6-methyladenosine (m6A) sites. N6-methyladenosine is a common and critical modification in eukaryotic mRNA, affecting various aspects of RNA metabolism. This includes stability, splicing, and translation. The prediction and detection of m6A sites are essential for understanding how this modification influences gene expression and cellular processes. In our work, we utilized datasets

from human, rat, and mouse tissues, specifically focusing on brain, kidney, and liver samples for each species.

The results for RNA N6-methyladenosine (m6A) site prediction in Table 4 clearly demonstrate that BiRNA-BERT outperforms RiNALMo and Deepm6A-MT across almost all evaluated tissues and species. In the human dataset, BiRNA-BERT shows the strongest performance in all three tissues, achieving 0.753 in brain (+3.15% over RiNALMo), 0.811 in kidney, and 0.816 in liver, both slightly higher than the second-best baselines. This trend continues in the rat dataset, where BiRNA-BERT reaches 0.788 in brain (+1.99%), 0.837 in kidney, and 0.825 in liver, demonstrating robust performance in tissues where RiNALMo previously performed well. In the mouse dataset, BiRNA-BERT again leads in brain (0.801) and kidney (0.820), while achieving a marginal edge in liver (0.747) over Deepm6A-MT. Statistically, the results are significant with a $p$ value of 0.009 between RiNALMo and BiRNA-BERT.

When averaged across tissues, BiRNA-BERT yields a 1.42% improvement in human, 1.10% in rat, and 0.84% in mouse, indicating strong cross-species generalization. The improvements are especially pronounced in brain tissues, which often contain more complex transcriptomic signatures. Even in cases where the numerical gain may seem small, such as human liver or mouse liver, BiRNA-BERT either outperforms or matches the highest baseline, doing so consistently across all settings.

## Nucleotide-level task

Along with sequence-level tasks described in the previous sections, BiRNA-BERT can simultaneously be applied to different nucleotide-level tasks where the embedding information per nucleotide is required. It is worth mentioning here that the dual tokenization scheme of BiRNA-BERT is not applicable in the case of such nucleotide-level tasks, as BPE compression will cause the model to lose information regarding particular nucleotide positions within the sequence. Thus, all the results provided here are based on the NUC level tokenization scheme of the BiRNA-BERT. However, due to the presence of length generalization with AliBi encoding in BiRNA-BERT, it provides an extra advantage in long sequence predictions even with NUC level tokenization, as we discuss in the following sections.

To verify BiRNA-BERT's capability of such granular-level tasks, we investigate the performance of BiRNA-BERT on three structural tasks for RNA: RNA 3D torsion angle prediction, RNA 3D distance map prediction, and RNA secondary structure prediction. Detailed description of the datasets along with the training and testing data is provided in the methods section.

**RNA secondary structure prediction.** The secondary structure of RNA is characterized by the presence of loops, particularly in the context of identifying open and closed regions within the sequence. This structural relationship is represented through a secondary structure matrix, which indicates whether pairs of nucleotides form a hydrogen bond. In RNA secondary structure prediction, the task is to predict the binding probability between nucleotide positions. To address this, we fine-tune BiRNA-BERT by incorporating a prediction head that forecasts the upper triangle of the secondary structure matrix, since the complementary positions within the matrix are symmetric. Here, we use a simple prediction head with a single-layer 2D convolutional network to avoid any architectural artifacts and focus solely on the inherent understanding capability of the language model itself.

In addition to BiRNA-BERT, we fine-tune RiNALMo and RNA-FM, employing the same prediction head architecture. Hyperparameter tuning is performed over both the learning rate and the number of epochs, with the validation dataset guiding the optimization process. Detailed descriptions of the model architecture and dataset are provided in the methods section.

As shown in Table 5, BiRNA-BERT outperforms both RiNALMo and RNA-FM in terms of F1-score, on both the validation dataset. In the test set, both BiRNA-BERT and RiNALMo provide a F1 score of 0.71, and a more substantial performance gain of 3.50% when compared to RNA-FM. Notably, the statistical significance test shows no significant difference between the results on RiNALMo and BiRNABERT ($p$ value = 0.08). Notably, RiNALMo is six times larger than BiRNA-BERT in terms of model parameters and utilized 22 times more computational resources for training, whereas RNA-FM's model configuration is comparable to that of BiRNA-BERT. We note that the results presented in Table 5 differ from the original figures reported in RiNALMo[6], as we employed a custom, simple CNN-based prediction head to provide a more direct comparison of the language models' inherent ability to capture RNA structural properties, rather than focusing exclusively on complex methods for RNA secondary structure prediction. Remarkably, U-Fold[26] which is a sophisticated method explicitly designed for RNA secondary structure prediction, provides a F1-score of

### Table 3 | Comparison of RNA-protein interaction prediction performance across different models

| Protein | Model | Accuracy | F1 | Recall |
|---------|-------|----------|-----|--------|
| AARS | BiRNA-BERT | **0.7053** | **0.7157** | 0.7638 |
| | RiNALMo | 0.6849 | 0.7092 | **0.7730** |
| | BERT-RBP | 0.6730 | 0.6797 | 0.6940 |
| | RNA-BERT | 0.6730 | 0.6710 | 0.6670 |
| AATF | BiRNA-BERT | **0.7501** | **0.7401** | 0.7407 |
| | RiNALMo | 0.7227 | 0.7322 | **0.7684** |
| | BERT-RBP | 0.7041 | 0.7016 | 0.6956 |
| | RNA-BERT | 0.6836 | 0.6886 | 0.6996 |
| ABCF1 | BiRNA-BERT | **0.7859** | **0.7860** | 0.8076 |
| | RiNALMo | 0.7500 | 0.7721 | **0.8442** |
| | BERT-RBP | 0.7180 | 0.7066 | 0.6793 |
| | RNA-BERT | 0.6866 | 0.7104 | 0.7686 |
| AGGF1 | BiRNA-BERT | **0.7162** | 0.7046 | 0.7023 |
| | RiNALMo | 0.7126 | **0.7219** | **0.7523** |
| | BERT-RBP | 0.6731 | 0.6861 | 0.7146 |
| | RNA-BERT | 0.6623 | 0.6829 | 0.7273 |
| AKAP1 | BiRNA-BERT | **0.7262** | 0.7224 | 0.7318 |
| | RiNALMo | 0.7037 | **0.7289** | **0.7975** |
| | BERT-RBP | 0.6775 | 0.6795 | 0.6840 |
| | RNA-BERT | 0.6528 | 0.6733 | 0.7156 |

Best results per metric are **bolded**, and second-best are underlined.

### Table 4 | Accuracy for RNA N6-methyladenosine site prediction on brain, kidney, and liver tissues across human, mouse, and rat organisms

| Species | Human | | | Mouse | | | Rat | | |
|---------|-------|------|-------|-------|------|-------|------|------|-------|
| Tissue | Brain | Kidney | Liver | Brain | Kidney | Liver | Brain | Kidney | Liver |
| BiRNA-BERT | **0.753** | **0.811** | **0.816** | **0.801** | **0.820** | **0.747** | **0.788** | **0.837** | **0.825** |
| RiNALMo | 0.7300 | 0.803 | 0.815 | 0.7961 | 0.8096 | 0.7435 | 0.7726 | 0.8341 | 0.8221 |
| Deepm6A-MT | 0.751 | 0.809 | 0.811 | 0.799 | 0.816 | 0.733 | 0.785 | 0.840 | 0.818 |

Results are shown for BiRNA, RiNALMo, and Deepm6A-MT. Best results are **bolded**, second-best are underlined.

**Table 5 | Summary of performance across different RNA structure prediction tasks**

| Method | 3D Torsion Angle (Mean Absolute Error) | | | | 3D Distance Map (R2 Score) | Secondary Structure (F1 Score) | |
|---|---|---|---|---|---|---|---|
| | VL | TS1 | TS2 | TS3 | Validation | Validation | TS0 |
| BiRNA-BERT | <u>**27.58**</u> | <u>**27.74**</u> | 26.22 | <u>31.45</u> | <u>**0.82**</u> | <u>**0.72**</u> | <u>**0.71**</u> |
| RNA-FM | 28.33 | 29.92 | 27.71 | 32.00 | 0.71 | 0.66 | 0.69 |
| RINALMo | 27.89 | 28.62 | <u>25.92</u> | 31.51 | 0.81 | 0.71 | <u>**0.71**</u> |
| SPOT-RNA-1D | 29.94 | 27.80 | **25.21** | **25.24** | – | – | – |
| U-Fold | – | – | – | – | – | – | 0.65 |

Best values are denoted in **bold**, second best in *italic*, and the best within the language models is underlined.

0.65 on the independent test dataset, wheras our BiRNA-BERT demonstrates superior performance with an F1-score of 0.71.

In Supplementary Fig. S3, we present the secondary structure prediction results for the longest sequence in the independent test dataset, which contains 441 nucleotides. For this complex sequence, which exhibits intricate loop structures, BiRNA-BERT significantly outperforms RiNALMo (F1-score of 0.98 for BiRNA-BERT versus 0.92 for RiNALMo). This result highlights BiRNA-BERT's robustness in maintaining high prediction accuracy even with longer sequences and complex structures, a capability enabled by its rotational positional encoding.

**RNA 3D distance map prediction.** The 3D (tertiary) structure of an RNA sequence is determined by the three-dimensional arrangement of the molecules that comprise the RNA. Specifically, the phosphate backbone forms the structural foundation of RNA, with nitrogenous bases and the sugar group positioned along the backbone according to their orientation. The first step in constructing the all-atom 3D structure of RNA involves determining the phosphate backbone of the RNA sequence. Predicting the 3D distance map of RNA's phosphate molecules from its sequence provides crucial inter-atomic distance constraints, enabling the construction of the absolute positions of the backbone atoms.

Similar to the RNA secondary structure prediction task discussed previously, 3D distance map prediction is achieved by fine-tuning the RNA language model with a specialized prediction head. This head predicts the inter-molecular phosphate atom distance for each pair of nucleotides. Unlike the binary prediction in secondary structure prediction (binding/non-binding), this prediction head provides a continuous regression value representing the distance between the phosphate atoms. A detailed description of the dataset used for this task is provided in the methods section. We curate a hold-out validation set following the approach in[27] to evaluate and compare the performance of BiRNA-BERT, RiNALMo, and RNA-FM.

As shown in Table 5, BiRNA-BERT and RiNALMo both significantly outperform RNA-FM in this task. BiRNA-BERT achieves a slightly higher $R^2$ score compared to RiNALMo, with an improvement of 1.24%. In Supplementary Fig. S4, the lengthwise performance analysis of BiRNA-BERT and RiNALMo for 3D distance map prediction task is presented. These results indicate that BiRNA-BERT shows consistency in performance in both short and long sequences.

**RNA torsion angle prediction.** RNA 3D torsion angles are critical parameters that define the three-dimensional conformation of RNA molecules. In 3D torsion angle prediction, we need to predict 7 torsion angles for each nucleotide in a sequence: $\alpha, \beta, \gamma, \delta, \epsilon, \zeta$, and $\chi$. These angles describe the rotations around the bonds that connect the nucleotides within an RNA strand, influencing its overall structure and stability. These are mathematically represented as the dihedral angles between four consecutive atoms in the RNA backbone.

During finetuning, the prediction head outputs pairwise sine and cosine values for each angle, resulting in 14 output nodes. This is a regression task, and we minimize the mean squared error (MSE) between the predicted and actual sine-cosine pairs. Dataset descriptions and finetuning details are provided in the methods section. We can see from Table 5, BiRNA-BERT outperforms RNA-FM in all the datasets including validation and three test sets. BiRNA-BERT performs better than RiNALMo in the validation dataset as well as two out of the three test sets. SPOT-RNA-1D[28] is a sophisticated method developed for RNA torsion angle prediction. In Table 5, we see that SPOT-RNA-1D outperforms the RNA language model-based approaches in TS2 and TS3 datasets. Note that in our comparative analysis of RNA language models for torsion angle prediction, we maintained a deliberately simple and generalizable prediction head to ensure a fair assessment of model capabilities. Despite this, BiRNA-BERT outperformed the specialized torsion angle prediction method SPOT-RNA-1D on both the validation and TS1 datasets, highlighting its robustness and effectiveness even without task-specific architectural enhancements.

**Information theoretic analysis: BPE tokenization comes with absolute information loss**

In this section, we estimate and compare the per-character information content of nucleotide (NUC) and BPE tokenizations using Shannon entropy. A nucleotide token, assuming a uniform distribution over A, T, C, and G, has a theoretical upper-bound entropy of 2 bits. Empirically, based on observed frequencies $P(A) \approx 0.2726$, $P(T) \approx 0.2144$, $P(C) \approx 0.2664$, and $P(G) \approx 0.2465$, the entropy of nucleotide tokens is

$$H_e(X_{NUC}) \approx 1.9939 \text{bits}.$$

For BPE tokens, we observe that their probabilities follow an exponential decay $P(x_i) = \frac{C}{2^{ai}}$ with best-fit parameters $C \approx 0.005086$ and $a \approx 0.011909$. This leads to the analytical expression for BPE entropy:

$$H(X_{BPE}) \approx \log_2\left(\frac{(C+1)^{(C+1)/C}}{C}\right).$$

Plugging in the empirical value of $C$ yields

$$H_e(X_{BPE}) \approx 9.1044 \text{bits}.$$

Given the average BPE token length $\bar{L} \approx 6.0768$, the per-character entropy of BPE is

$$\hat{H}_e(X_{BPE}) = \frac{H_e(X_{BPE})}{\bar{L}} \approx \frac{9.1044}{6.0768} \approx 1.498.$$

Comparing this with the nucleotide entropy, we obtain the empirical entropy ratio:

$$\frac{\hat{H}_e(X_{BPE})}{\hat{H}_e(X_{NUC})} = \frac{1.498}{1.9939} \approx 0.7514 < 1.$$

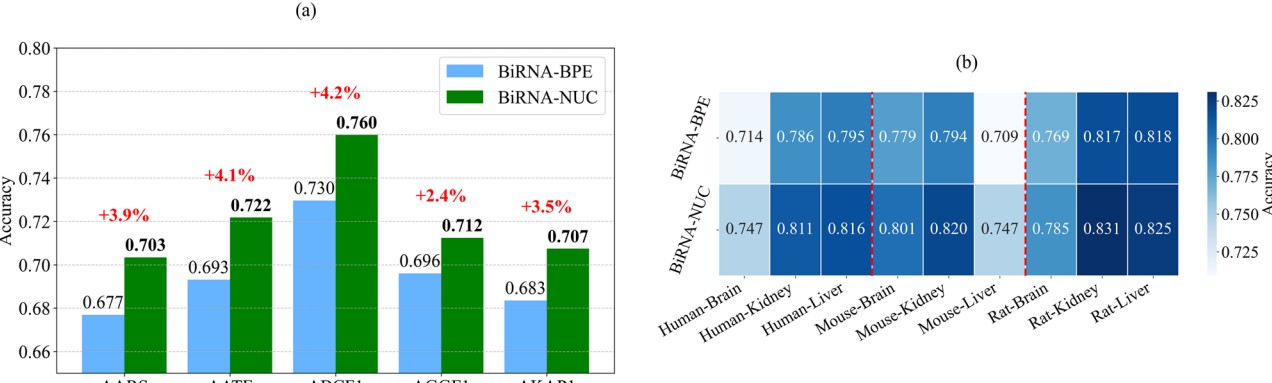

**Fig. 4 | Comparison between BiRNA-BPE and BiRNA-NUC tokenization. a** Improvement on BiRNA-BERT (NUC) over BiRNA-BERT (BPE) in RNA-Protein interaction across five protein families. **b** Comparison of the NUC and BPE tokenization on RNA N6-methyladenosine site prediction across different species and tissues.

This indicates that BPE tokenization results in roughly 25% lower per-character information content than NUC. While this reflects a loss in granularity, it enables up to 6x longer sequences to be processed with the same memory budget, making BPE advantageous for long-sequence modeling. The reduced entropy also helps explain the observed performance gap between BPE and NUC on short-sequence tasks. Detailed derivation of the entropy based information content analysis is provided in the supplementary file.

### NUC tokenization is preferable over BPE if memory allows

We have shown that BPE tokenization incurs information loss, with each BPE token containing only 75% of the information content compared to nucleotide (NUC) tokens, while simultaneously reducing the token count by 6-fold. In this section, we show that for sequences that fit within the maximum length constraint of the RNA language model, NUC tokenization is preferable over BPE tokenization. We validate this information-theoretic finding through empirical evaluation on short sequences in two downstream binary classification tasks: RNA-Protein interaction prediction (sequence length of 101), and RNA N6-methyladenosine site prediction (sequence length of 41).

In Fig. 4a, we present results on RNA-Protein interaction prediction for BPE and NUC tokenization under two different configurations of BiRNA-BERT. NUC consistently outperforms BPE on all five concerned datasets by 3.91% (AARS), 4.10% (AATF), 4.24% (ABCF1), 2.36% (AGGF1), and 3.50% (AKAP1).

Results for RNA N6-methyladenosine site prediction are presented in Fig. 4b. Similar to the RNA-Protein interaction task, NUC versions perform consistently better than BPE versions. The average improvement of NUC over BPE for human, mouse, and rat is 3.88%, 2.24%, and 1.68% in three different tissues: brain, kidney, and liver.

These results suggest that, for short sequence tasks where the sequence lengths fit within the maximum context window of the model, NUC tokenization provides better fidelity and downstream performance.

### When to use which tokenization scheme?.

- *Use NUC tokenization* when sequence lengths fit comfortably within the model's maximum token limit and GPU memory permits. This is ideal for short or moderate-length sequences in classification and binding prediction tasks.
- *Use BPE tokenization* when working with long RNA sequences that would otherwise exceed the model's input length or exhaust memory. BPE significantly reduces sequence length while maintaining an acceptable performance trade-off.Refer to the official GitHub repository at github.com/buetnlpbio/BiRNA-BERT for implementation details, example scripts, and utilities to run BiRNA-BERT under both NUC and BPE tokenization schemes.

### Empirical perplexity analysis of different RNA language models

The information-theoretic analysis of NUC and BPE tokenization in the previous section reveals that BPE tokens preserve approximately 75% of the information contained within the NUC tokens. Building on this, we now compare the models' inherent ability to reconstruct masked tokens and to model the underlying token-level probability distributions.

Although perplexity is traditionally defined for autoregressive generative models, we adopt an alternative definition suited for masked language models (MLMs), as proposed in prior biological language modeling work[15]. In this setting, the perplexity is computed over the masked positions by taking the exponential of the average loss:

$$\text{perplexity}(\theta, s) = 2^{l(\theta, s)}, \tag{2}$$

where $l(\theta, s)$ denotes the average log loss over the masked tokens.

To formalize, given a nucleotide sequence $s$ of interest, a set of masked positions $\mathcal{P}_{\text{masked}}$ is selected either by masking the central token or by randomly masking a fixed percentage (e.g., 15%) of tokens. The masked sequence is fed into the model to predict probabilities over the vocabulary $\mathcal{V}$ at each masked site. The loss function and accuracy are defined as

$$l(\theta, s) = \sum_{i \in \mathcal{P}_{\text{masked}}} \sum_{\text{tok} \in \mathcal{V}} \log p(\theta, i, \text{tok}) \cdot \mathbf{1}(\text{tok} = s(i)),$$

$$acc(\theta, s) = \frac{1}{|\mathcal{P}_{\text{masked}}|} \sum_{i \in \mathcal{P}_{\text{masked}}} \mathbf{1}\left(argmax_{\text{tok} \in \mathcal{V}} \log p(\theta, i, \text{tok}) = s(i)\right), \tag{3}$$

We consider both the approach (central token masking and random 15% token masking) to compare the recovery accuracy and perplexity for different version of BiRNA-BERT, RiNALMo, and RNA-FM. For BiRNA-BERT, RiNALMo, and RNA-FM, we use a curated dataset from the rfam families with 100k sequences.

Supplementary Table S4 shows the perplexity and recovery accuracy for BPE-based models (BiRNA-BERT (BPE)), as well as NUC-based models (BIRNA-BERT (NUC), RNA-FM, and RiNALMo). We see that the BPE-based models have significantly lower recovery accuracy compared to the NUC-based models. This is due to the fact that in BPE-based models the vocabulary size is ~4000 while in NUC-based models the primary vocabulary consists of only four nucleotides along with some additional special characters. The probability of being correct by chance for BPE-based models is $\sim \frac{1}{4000}$ while this is $\sim \frac{1}{4}$ for NUC-based models. Thus in BPE-based models, the recovery is significantly tougher compared to the NUC-based models.

RiNALMo and BiRNA-BERT(NUC) both provide equivalent perplexity in central as well as random 15% masking scheme, while RiNALMo

performs slightly better in terms of recovery accuracy (1.2% ↑). RNA-FM provides significantly higher perplexity (3.3% ↑) compared to both RiNALMo and BiRNA-BERT, which indicates that, in terms of pretraining, RNA-FM lags behind BiRNA-BERT and RiNALMo. This similarity in perplexity and token recovery accuracy suggests that BiRNA-BERT and RiNALMo acquire a comparable understanding of RNA sequence patterns during pretraining, despite notable differences in architecture–BiRNA-BERT is approximately six times smaller and adopts a dual-tokenization strategy, whereas RiNALMo uses single nucleotide-level tokenization. The downstream task performances further support this, as BiRNA-BERT matches RiNALMo across several benchmarks while offering faster inference. This indicates that BiRNA-BERT does not suffer from any inherent limitations in RNA language understanding, and in fact, provides additional advantages—particularly its ability to process long sequences efficiently through adaptive tokenization.

### Comparison of computational efficiency

In this section, we compare the pretraining computational resources across different RNA language models, as well as present a comparative analysis of the inference and finetuning times for those models on different downstream tasks.

**Computational resources for pretraining.** To provide a fair perspective on the computational efficiency of different RNA language models, we summarize the hardware configurations, training durations, and theoretical peak FP16 performance of the GPUs used for pretraining RiNALMo and BiRNA-BERT. We also compute the relative resource usage ratios based on the number of GPUs, their peak throughput, and the total training time. This comparison highlights the lightweight nature and faster pretraining requirements of BiRNA-BERT relative to existing large-scale models.

RiNALMo. RiNALMo utilizes **7 GPUs** of the `A100` (80GB). The total memory usage is calculated as 7 × 80GB = 560GB. The model requires **14 days** for training, resulting in a total training time of 336 hours.

BiRNA-BERT. BiRNA-BERT is trained using **8 GPUs** of the `3090` (24GB). The total memory usage for BiRNA-BERT is 8 × 24GB = 192GB. Despite using fewer GPUs and less total memory, BiRNA-BERT completes training in **48.42 hours**.

In Supplementary Fig. S5, the total usage of GPU memory and the training time are shown for the RNA language models (BiRNA-BERT and RiNALMo). In each setting, we see that dual tokenization and rotational positional embedding-based foundation models require significantly lower computational resources.

We trained *BiRNA-BERT*, our 116M parameter model, for two epochs over the same dataset used by RiNALMo[29], to ensure consistency and enable a fair comparison. The training was carried out using 8 NVIDIA RTX 3090 GPUs (24GB each), requiring a total of approximately 90 hours. In contrast, RiNALMo, a 600M parameter model, was trained over 14 days (336 hours) on 7 A100 GPUs (80GB each). Given that the training time per step scales approximately linearly with the parameter count[30], the expected speed-up from model size alone is $\frac{600M}{116M} \approx 5.17$. Accounting for hardware differences, where A100 GPUs provide about 2.25 × the throughput of RTX 3090s[31], the hardware-adjusted scaling factor is $\frac{8}{7 \times 2.25} \approx 0.51$. Additionally, we incorporated FlashAttention[32], which yields a 2–4 × speedup in attention computation. As a result, our training process benefits from a conservative 1.5 × overall acceleration. Therefore, the total expected acceleration is 5.17 × 0.51 × 1.5 ≈ 3.95, resulting in an estimated training time of $\frac{336}{3.95} \approx 85$ hours—well aligned with our observed 90-hour training duration.

**Inference and finetuning time.** This section compares the inference time and finetuning time of different RNA language models. Supplementary Fig. S6a presents the inference time for a single sequence across varying lengths, showing that while RiNALMo and RNA-FM encounter out-of-memory errors for sequences longer than 1024 nucleotides, BiRNA-BERT consistently provides embeddings even for sequences exceeding this length. Supplementary Fig. S6b compares the finetuning time per epoch for three downstream tasks: miRNA-lncRNA interaction, RNA-protein interaction, and secondary structure prediction. The results indicate that BiRNA-BERT requires significantly lower training time per epoch compared to the other models.

## Discussion

The application of NLP techniques to biological research has seen renewed interest following the success of AlphaFold[33] in predicting protein structures. While the majority of research efforts have focused on protein sequences with considerable success, recent interest has begun to shift towards modeling DNA and RNA sequences.

In this work, we have introduced BiRNA-BERT, a compact Transformer encoder for RNA that combines adaptive dual tokenization with ALiBi positional encoding to overcome the limitations of existing language models. By integrating nucleotide-level and BPE tokenization during pretraining, BiRNA-BERT can flexibly process both short and arbitrarily long sequences without truncation, while still supporting fine-grained, per-nucleotide predictions. Despite using fewer parameters and far less compute than larger models, BiRNA-BERT matches or exceeds their performance across a variety of downstream tasks: long-sequence miRNA-lncRNA interaction, short-sequence RNA-protein binding and m6A site prediction, and nucleotide-level structural predictions such as secondary structure, distance maps, and torsion angles.

Our information-theoretic analysis clarifies how BPE compression trades off some per-character entropy for the ability to encode longer contexts, and our experiments show that—when memory permits—nucleotide-level tokenization still yields the best results on short sequences.

Looking ahead, there are several promising directions. For instance, Integrating our model into pipelines for single nucleotide polymorphism scoring, allele-specific binding prediction or RNA design would test its utility in real-world genomics and therapeutics. A more comprehensive evaluation in unsupervised settings—such as variant effect prediction or mutation impact analysis—could reveal new biological insights without task-specific training. Moreover, training larger versions of our dual-token model, or extending the approach to pan-species DNA and protein corpora, may further improve performance and broaden applicability. By uniting flexible tokenization with efficient training, BiRNA-BERT opens the door to practical, high-accuracy language modeling across the full spectrum of nucleic acid research. We hope this work encourages further exploration of adaptive tokenization strategies and their impact on biological sequence understanding.

## Methods

The schematic diagram of BiRNA-BERT is shown in Fig. 5, highlighting the major challenges in RNA language modeling and demonstrating how BiRNA-BERT addresses these challenges. We describe the key components and training strategies of BiRNA-BERT in the subsequent sections. We first describe the motivation behind using relative positional encoding (ALiBi) for extrapolating the trained model to longer sequences in downstream tasks (the section "Positional encoding in the transformer architecture"). Then we discuss the tokenization strategies for biological foundation models and the applicability of using BPE tokenization for BiRNA-BERT (the section "Tokenization strategies for biological foundational models"). In the section "Adaptive tokenization and dual pretraining", we discuss the dual tokenization pretraining approach and its motivation. Finally, we describe the pretraining configurations and datasets used in the sections "Pretraining configuration" and "Pretraining dataset".

### Positional encoding in the transformer architecture

Since the attention mechanism is permutation-invariant, positional information must be explicitly added. There are two main strategies for encoding positional information—fixed and relative. In fixed positional

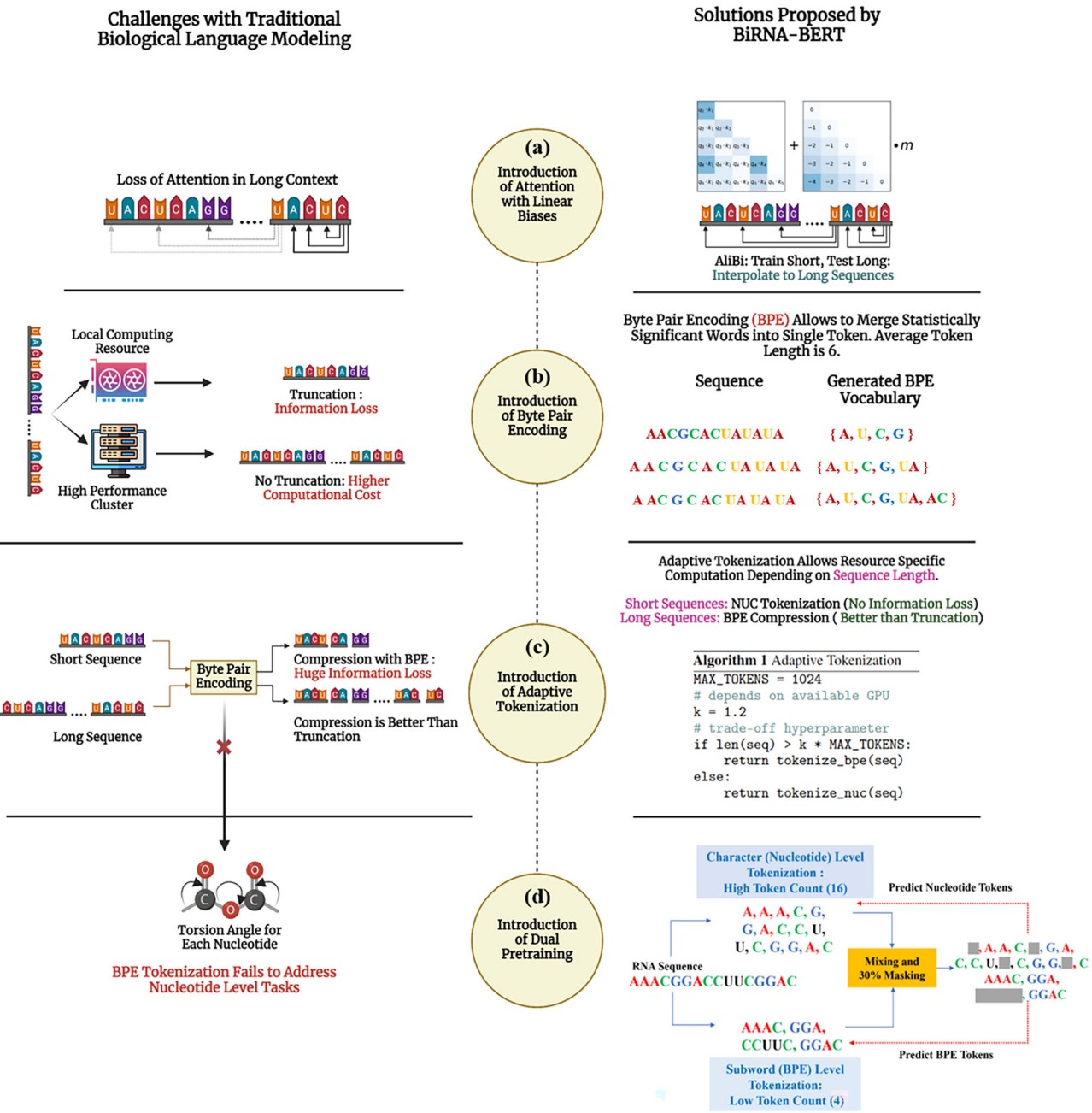

**Fig. 5 | Schematic diagram of the overall architecture of BiRNA-BERT, illustrating its components designed to address key challenges in RNA language modeling. a** BiRNA-BERT resolves the issue of loss of attention in long sequences by incorporating AliBi positional encoding. **b** For low-compute environments where large sequences cannot fit into memory, BiRNA-BERT employs BPE tokenization, which merges statistically significant residues into single tokens, allowing for longer context lengths within limited resources. **c** Although BPE compression is more effective than sequence truncation for long sequences, it leads to information loss and poor performance in shorter sequences. To address this, BiRNA-BERT uses adaptive tokenization: NUC tokenization is preferred for short sequences, while BPE tokenization is applied to long sequences, avoiding truncation. **d** To enable residue-level downstream tasks, BiRNA-BERT introduces dual pretraining, supporting both sequence-level and residue-level tasks simultaneously. The figure was created with BioRender.com.

encoding schemes such as sinusoidal[34] or learned embeddings[1], the positional information is a vector function of the position index within some predefined context length. Since the positional information is explicit in the form of a vector, these methods cannot extrapolate to context lengths beyond those seen in pretraining. Previous biological foundation models[3,5] have mostly used fixed positional embeddings following the original BERT implementation. As a result, long sequences had to be truncated (usually to 1024 tokens) on downstream tasks. To overcome this challenge, we leverage relative positional encoding as shown in Fig. 5a. In relative positional encoding, the positional information is a function of the distance between two tokens. Popular algorithms for relative positional embeddings are T5 Bias[35], Rotary Positional Embedding (RoPE)[36], and Attention with Linear Biases (ALiBi)[11]. Recently, RiNALMo[6] utilized Rotary Positional Embedding (RoPE) but still truncated sequences to 1022 nucleotides since extending the context window using RoPE requires additional training to preserve performance. DNABERT-2[4] was the first biological foundation model to use ALiBi, which when combined with BPE, allowed them to process sequences of up to 10000 nucleotides, significantly outperforming recurrent models such as HyenaDNA[37] on long-sequence tasks.

**Attention with linear biases (ALiBi)**. In contrast to complex methods such as T5 Bias and RoPE which are hard for models to extrapolate without continued pretraining, ALiBi simply reduces the attention score between two tokens by a scaler function of their distance.

Let $\mathbf{Q} \in \mathbb{R}^{L \times d}$, $\mathbf{K} \in \mathbb{R}^{L \times d}$, and $\mathbf{V} \in \mathbb{R}^{L \times d}$ represent the query, key, and value matrices, where $L$ is the sequence length and $d$ is the hidden dimension. ALiBi modifies the attention scores by adding a bias term, a function of the positional distance between the query and key elements. Specifically, for positions $i$ and $j$, the linear bias term $b_{ij}$ is defined as $b_{ij} = \alpha \cdot |i - j|$, where $\alpha$ is a hyperparameter. This results in modified attention scores $\mathbf{A}_{ij} = \frac{\mathbf{Q}_i \mathbf{K}_j^\top}{\sqrt{d}} - b_{ij}$. Since ALiBi explicitly penalized the attention scores between distance tokens, the $\alpha$ values must be carefully chosen to preserve long-context capabilities. The authors of ALiBi use different $\alpha$ values for different attention heads, uniformly sampled from the geometric progression $\frac{1}{2}, \frac{1}{2^2}, ..., \frac{1}{2^8}$. Intuitively, some attention heads specialize in aggregating local information while others preserve long-context information.

### Tokenization strategies for biological foundational models

Nucleotide-level tokenization (NUC) has long been the dominant form of tokenization in biological sequence modeling[5,6], partially due to the need for NUC embeddings in secondary tasks for RNA and protein. An alternative form of tokenization is $k$-mer tokenization, where segments of $k$ nucleotides are considered as one token. The motivation behind $k$-mer tokenization is to effectively capture the biological significance of RNA sequences, as certain $k$-mers, such as 3-mers (codons), directly correspond to amino acids, which are the building blocks of proteins. DNABERT[3] uses overlapping $k$-mers while models such as Nucleotide Transformers[15] use non-overlapping k-mers. However, these full sequence tokenization schemes are constarined by the memory limitation. Even with relative positional encoding, it is challenging to fit longer context length sequences into available memory. To tackle this challenge, we have employed Byte Pair Encoding (BPE) tokenization as shown in Fig. 5b. BPE[13,14] is a subword tokenization technique that iteratively merges the most frequent pairs of bytes or characters to create new tokens, thereby reducing the vocabulary size to a fixed number.

Recently, DNABERT-2[4] demonstrated the superiority of BPE tokenization over NUC and $k$-mer tokenization for DNA sequences. Since DNA lacks notable downstream tasks that necessitate NUC embeddings, BPE tokenization proved to be the most suitable approach for this context.

**Example of byte-pair encoding (BPE) on RNA sequences**. Let us consider a toy RNA sequence:

$$S = \text{AUGGCUACUGCAUGCUAGUCA}$$

1. *Initial vocabulary*: {A, U, G, C} The sequence is first split into individual characters:

   [A, U, G, G, C, U, A, C, U, G, C, A, U, G, C, U, A, G, U, C, A]

2. *Step 1—Merge most frequent pair*, e.g., AU → AU After merging all instances of A followed by U, the sequence becomes:

   [AU, G, G, C, U, AU, C, U, G, C, AU, G, C, U, AU, G, U, C, A]

3. *Step 2—Merge next frequent pair*, e.g., GC → GC The sequence is updated as

   [AU, G, G, GC, U, AU, C, U, GC, AU, G, C, U, AU, G, U, C, A]

4. *Step 3—Continue merging*, e.g., UG → UG The sequence becomes:

   [AU, G, G, GC, U, AU, C, UG, C, AU, G, C, U, AU, UG, U, C, A]

5. *Final tokenized output (illustrative)*: After several merge steps, the sequence might be tokenized as

   $$\text{Tokens} = [\text{AU, GG, CU, AC, UG, CA, UG, CUA, GUC, A}]$$

This example demonstrates how BPE progressively merges frequent subsequences to form longer and more semantically meaningful tokens, effectively balancing compression and biological relevance in RNA sequence modeling.

### Adaptive tokenization and dual pretraining

BPE tokenization performs well for longer sequences, offering a significant advantage over fixed-length truncation, as discussed in the results section. However, for shorter sequences, this compression method can result in information loss (the section "Information theoretic analysis: BPE tokenization comes with absolute information loss"). To address this, we implemented an adaptive tokenization strategy: BPE tokenization is employed for long sequences, while NUC (nucleotide-level) tokenization is used for shorter ones, as illustrated in Fig. 5c. Another limitation of BPE tokenization is its inability to handle nucleotide-level tasks, such as torsion angle and secondary structure prediction, as BPE tokenization compresses multiple nucleotides into a single token. To overcome this, we propose a dual pretraining approach as shown in Fig. 5d, which allows a single model to learn both BPE tokens and NUC tokens effectively.

The primary motivation behind training a single model with dual tokenization is memory efficiency. Although having separate BPE and NUC models is viable, we argue that a single model training on both is memory-efficient since only one set of weights needs to be on memory and can process longer sequences and larger batch sizes. Memory efficiency is crucial since the transformer architecture has $\mathcal{O}(n^2)$ memory requirement which especially penalizes long sequence tasks. In addition to BiRNA-BERT, We train two more models BPE-Only-BERT and NUC-Only-BERT to show that dual tokenization does not affect downstream performance. Performing BPE tokenization on long sequences effectively compresses the sequence with average BPE token length $\bar{L} \approx 6.0768$. We show in the section "Information theoretic analysis: BPE tokenization comes with absolute information loss" that the empirical per-character entropy ratio is

$$\frac{\hat{H}_e(X_{BPE})}{\hat{H}_e(X_{NUC})} \approx 0.7514 < 1$$

where $\hat{H}_e(X_{BPE})$ is the average character-level entropy of the BPE representation of a sequence and $\hat{H}_e(X_{NUC})$ is the per-character entropy for nucleotide tokenization. Therefore, BPE tokenization is essentially a trade-off between information compression and computational efficiency,

Given an RNA sequence $\mathbf{r} = (r_1, r_2, ..., r_n)$, we apply two types of tokenization: BPE and NUC.

- *BPE Tokenization*: The BPE tokenized sequence is $\mathbf{r}_{BPE} = (\beta_1, \beta_2, ..., \beta_m)$ with a vocabulary size of 4096.
- *Nucleotide-Level Tokenization*: The nucleotide tokenized sequence is $\mathbf{r}_{Nuc} = (v_1, v_2, ..., v_n)$ with a vocabulary size of 4.

We randomly mask tokens in both $\mathbf{r}_{BPE}$ and $\mathbf{r}_{Nuc}$ for Masked Language Modeling (MLM) training:

- *Masked BPE Sequence*: $\mathbf{r}_{BPE}^{mask}$ where a subset of $\beta_i$ is replaced with a mask token [MASK].
- *Masked Nucleotide Sequence*: $\mathbf{r}_{Nuc}^{mask}$ where a subset of $v_i$ is replaced with a mask token [MASK].

The total loss $\mathcal{L}_{total}$ is the sum of the BPE and nucleotide losses:

$$\mathcal{L}_{total} = -\sum_{i \in \mathcal{M}_{BPE}} \log P(\beta_i | \mathbf{r}_{BPE}^{mask}) - \sum_{i \in \mathcal{M}_{Nuc}} \log P(\nu_i | \mathbf{r}_{Nuc}^{mask}).$$

The parameters of the BERT model, $\Theta$, are optimized to minimize the total loss:

$$\Theta^* = \arg\min_{\Theta} \mathcal{L}_{total}.$$

## Pretraining configuration

We use the MosaicML framework[9] for efficient training. We pretrain all models with a learning rate (LR) of $2 \times 10^{-4}$, a warmup ratio of 0.06, a linear rate scheduler that decreases to 0.02 of the starting LR, and a batch size of 200 per device, totaling 1600 samples per batch across eight Nvidia RTX 4090 GPUs. In total, 26.42 billion tokens, including NUC and BPE tokens, are used for training the BiRNA-BERT language model. We pretrain the model for 2 complete epochs.

## Pretraining dataset

To assemble a robust dataset of ncRNA sequences, we utilized the RNA-central database[16]. RNAcentral serves as an exhaustive repository, aggregating data from multiple ncRNA databases, thereby providing a wide array of ncRNA sequences. The dataset from RNAcentral encompasses approximately 36 million RNA sequences, collectively containing 26.42 billion nucleotides. This extensive compilation allows for a thorough representation of the ncRNA landscape to facilitate effective model training.

## Downstream tasks and datasets

To evaluate the performance of BiRNA-BERT under different configurations, we consider different downstream tasks across various length categories. In this section, we will focus on the datasets that are used for downstream tasks.

**miRNA-lncRNA interaction prediction.** Long non-coding RNA (lncRNA) related studies frequently involve much longer sequences which necessitated task-specific architectures such as PmliPred[38] and CORAIN[25]. We use three benchmarking datasets for the RNA-RNA interaction prediction task compiled by[38] (dataset details shown in supplementary Table S1. For evaluation, one benchmarking dataset is used as the training set, and another dataset is used for validation following the strategy used in[25]. Thus, we have 6 train-test combinations and we report performance in all these combinations. The lengths of the sequences in the miRNA dataset are 10 to 50, whereas the lncRNA dataset ranges from 200 to 4000. The length distributions of sequences for the miRNA-lncRNA Interaction Prediction task are shown in Supplementary Fig. S2. The training and testing are always performed in a cross-species manner following[25] which ensures that the sequence distributions are not biased by the training set.

**RNA–protein interaction prediction.** This focuses on finding the binding sites and interactions between RNA molecules and proteins to understand post-transcriptional regulation. Benchmark dataset for RNA-protein interaction prediction is collected from RBPsuit available at http://www.csbio.sjtu.edu.cn/bioinf/RBPsuite/. This database contains datasets for 154 different proteins and a collection of interacting and non-interacting RNA sequences for each protein. We consider a subset of 5 datasets for our evaluation (AARS, AATF, AKAP1, AGGF1, ABCF1). The length of RNA sequences used for this task is 101 across all the datasets. The number of sequences used for training, validation, and testing is shown in Supplementary Table S2. RBPsuit provides separate testing data with each RNA-binding protein dataset, ensuring that training and testing datasets are not within 80% sequence similarity threshold.

**RNA N6-methyladenosine prediction.** N6-methyladenosine (m6A) is a common and critical modification in eukaryotic mRNA, affecting various aspects of RNA metabolism. This includes stability, splicing, and translation. The prediction and detection of m6A sites are essential for understanding how this modification influences gene expression and cellular processes. In our work, we utilized datasets from human, rat, and mouse tissues, specifically focusing on brain, kidney, and liver samples for each species. Data sets were derived from the iRNA-m6A study available at http://www.biolscience.cn/Deepm6A-MT/data/, employing an antibody-independent m6A-REF-seq protocol, which is both high-throughput and accurate for m6A detection. Positive samples were selected based on the presence of m6A at the center of 41 continuous nucleotide residues, while negative samples were randomly selected from the same tissues but without m6A sites. CD-Hit removes the sequences that are within 80% similarity threshold between the training and the test set. The length of the sequences across all the datasets is 41. Dataset specifications are shown in Supplementary Table S3.

**RNA secondary structure prediction (nucleotide level task).** Accurate secondary structure prediction is pivotal for elucidating the structural and functional dynamics of RNA. To evaluate the performance of BiRNA-BERT in this task, we use the bpRNA[39] database, which is a large-scale collection of secondary RNA structures with detailed annotations of base-pairing patterns. The version of bpRNA that we use is taken from[27] which consists of 10,814 training structures (TR0), 1300 validation structures (VL), and 1305 testing structures (TS0). This version also ensures that CD-HIT is conducted with 90% similarity threshold.

**RNA 3D distance map prediction (nucleotide level task).** RNA 3D distance map estimates the physical distances between pairs of nucleotides within an RNA molecule. Following[40], we utilize a dataset derived from non-redundant RNA 3D structures documented by[41]. We perform 20% hold out validation test on the same validation data for all models considered.

**RNA 3D torsion angle prediction (nucleotide level task).** There are seven torsion angles commonly referred to as $\alpha$, $\beta$, $\gamma$, $\delta$, $\epsilon$, $\zeta$, and $\chi$. These angles describe the rotations around the bonds that connect the nucleotides within an RNA strand, influencing its overall structure and stability. These angles are mathematically represented as the dihedral angles between four consecutive atoms in the RNA backbone. For example, the $\alpha$ angle is measured as the dihedral angle between O5'-P-O3'-C3'. The dataset for RNA torsion angle prediction is collected from https://sparks-lab.org/server/spot-rna-1d/. The training (TR), validation (VL), and three test sets (TS1, TS2, and TS3) have 286, 30, 63, 30, and 54 RNA chains, with average sequence lengths of 122, 15, 30, 14, and 24 respectively. Particularly, according to ref. 28 the noncluster RNA sequences obtained after CD-HIT-EST and BLAST-N processing were randomly divided into one validation (VL) and two test sets (TS1 and TS2). In addition, for curating the TS3 dataset, the authors downloaded all the NMR structures (707 RNAs) from the PDB on April 5, 2021. After removing redundancy within TS3 and from all other RNA structures (TR, VL, TS1, and TS2) using the exact same specifications as TS1, 54 nonredundant RNA chains are obtained in TS3.

## Statistics and reproducibility

Statistical significance for downstream task results was assessed by computing p-values between the performance of BiRNA-BERT and that of the corresponding benchmark methods. A p-value below 0.05 was considered indicative of a statistically significant difference. To facilitate reproducibility, all necessary downstream neural network prediction heads, pretrained model checkpoints, and instructions for using the different tokenization modes are available in our GitHub repository at https://github.com/buetnlpbio/BiRNA-BERT.

## Reporting summary

Further information on research design is available in the Nature Portfolio Reporting Summary linked to this article.

## Data availability

The datasets underlying this article were derived from sources in the public domain.

## Code availability

The pretraining configuration, embedding generation, and finetuning codes are available in ref. 42.

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

## Acknowledgements
This work was partially supported by the Basic Research Grant to Md Shamsuzzoha Bayzid at Bangladesh University of Engineering and Technology (BUET).

## Author contributions
Md Toki Tahmid and Md. Shamsuzzoha Bayzid conceptualized the study and designed the experiments. Haz Sameen Shahgir performed the pretraining of the language model. Md Toki Tahmid performed the downstream result analysis. Sazan Mahbub performed the information-theoretic analysis. All authors participated in preparing the first draft and Md Toki Tahmid and Md. Shamsuzzoha Bayzid prepared the final draft. Yue Dong and Md. Shamsuzzoha Bayzid supervised the study.

## Competing interests
The authors declare no competing interests.
