## [Transparent Peer Review file · Communications Biology]

BiRNA-BERT allows efficient RNA language modeling with adaptive tokenization

Corresponding Author: Dr Md Shamsuzzoha Bayzid

Version 0:

Reviewer comments:

Reviewer #1

(Remarks to the Author)

Tahmid et al developed a new RNA model. This employed both nucleotide-level and adapting byte pair encoding that allows efficient calculations. Here are the comments.

- 1) Databases for training and test employed in the downstream tasks are not clearly described. They were mixed in the result section. Please place them in the method section.
- 2) There is no comparison regarding the benefit for introducing NUC and BPE encoding except on RNA-protein interaction prediction. Please add the method variants with NUC only and BPE only for unsupervised validation (such as clustering) as well as other downstream tasks.
- 3) All downstream tasks are sequence based. It is not clear if the LM can be useful for predicting secondary structure.
- 4) It is very strange that BiRNA-Bert is only comparing to RINALMO and RNA-FM for clustering. Please add some more recent RNA LM models. Also why 9 RNA structural families and 30 RFam families were chosen? It seems that 30 RFam families may have overlaps with 9 RNA structural families. Please use more RFam and more structures for clustering.
- 5) Why RINALMO is so bad for 9 structural families and so good for RFam families? This suggests that something is wrong with the datasets.
- 6) For miRNA-lncRNA interaction prediction, there are many methods. Can the authors compare with other methods in addition to CORAIN. Regarding the datasets, is there an overlap (>80% sequence ID) between training and test? This is biased or overestimating the performance.
- 7) For RNA splicing sites, please also check if there is overlap (>80% sequence ID) between training and test?
- 8) For torsion angles, please also compare to the results of SPOT-RNA-1D.
- 9) There are other tasks (N6-methyladenosine site, RNA-protein binding, promoter site, transcription factor). In all these tasks, the description of training, validation and test sets are very poor. It is not clear if >80% sequence ID between training, validation and test sets were removed.

Reviewer #2

(Remarks to the Author)

The manuscript proposes BiRNA-BERT, a BERT-style pre-trained on ncRNA and mRNA, to cope with long DNA/RNA sequences without sacrificing ability for processing nucleotide-level tasks. The key technique of the BiRNA-BERT is an adaptive dual tokenization scheme, which utilizes nucleotide-level (NUC) tokenization for short sequences and efficient BPE tokenization for long sequences. The results show that BiRNA achieves state-of-the-art results in long-sequence downstream tasks and achieves a performance comparable to 6x larger in short-sequence tasks. Besides, they also attempted to adapt their scheme to the DNA field, resulting in a BiDNA-BERT.

****Overview about the idea****

I think the core idea of BiRNA is very naive, clear, and straightforward. It indeed solves the long sequence problem to some extent by constructing a piecewise function that chooses different tokenization ways for sequences of different lengths. However, this way of combining the two tokenization methods is too simple for the selection only depends on the length. It leads to two suites of vocabularies generated by NUC and BPE tokenization, which are ponderously injected into one without any compatible means of fusion. Although the results show that the introduction of BPE will not affect the performance of the original NUC part of the model, it may just be because the is enough. Therefore, we need more analyses regarding the relationship between the NUC and BPE part in the BiRNA, instead of inspecting the performance of these two

parts. As for the DNA version adaptation of the adaptive tokenization, they did not demonstrate the main characteristic for long sequences, which can not become powerful evidence for their generalization ability. The specifics are expanded upon below:

1. **Task Composition** In the introduction, the authors mention that structure prediction is an important task for the nucleotide level, but why finally choose RNA 3D torsion angle prediction as the only evaluation at the nucleotide level? By the way, there are 4 sequence-level tasks but only 1 nucleotide-level task in this manuscript, and only the interaction task involves sequences longer than 1.2×10^4 (the threshold triggers BPE). In a word, the task composition is not well-balanced.

2. **Model Training** In section 2.4, the author asserts that their are trained for only one epoch to delve into the essence of their work. However, I believe that the lacks sufficient training for practical application. In training, the crucial juncture is the final checkpoint, representing the pinnacle of achievable performance. Moreover, monitoring loss variations throughout the training process can elucidate the complexity of the learning task.

3. **Model Inference** If you pre-trained your with adaptive tokenization, why is there a lack of an adaptive scheme for inference, relying solely on a single type of tokenization instead? (BiRNA-NUC and BiRNA-BPE) I am perplexed by the utilization of BiRNA-BERT, which appears to constrain its potential benefits.

3. **Dataset Composition** Why did the authors incorporate 0.5 million mRNA sequences alongside 36 million ncRNA sequences? The addition of mRNA seems to complicate the dataset and render attribution more challenging. While they suggest that these sequences can provide a rich source of coding RNA data for training, each mRNA sequence can only represent a single sample in a batch, constituting a small fraction of the training set. Furthermore, given that mRNA sequences are generally longer than ncRNA sequences on average, this may introduce a bias in BPE selection and training towards mRNA rather than ncRNA.

4. **Unsupervised Clustering** Why were the specific 9 RNA structural families—16s, 23s, 5s, RNaseP, grp1, srp, tRNA, telomerase, tmRNA—chosen? What distinguishes these "structural families" from the families in the Rfam database? The author also noted, 'However, the secondary structural family dataset was not explicitly included in the pretraining database.' Where were these data sourced from, and how were they ensured to be part of the pretraining dataset? It may be beneficial to include some indicators elucidating their relationship with RNACentral (the pretraining data.)

5. **Long-Sequence Task: miRNA-lncRNA Interaction Prediction** Given the considerable length of lncRNA, it is unsurprising that BiRNA-BPE surpasses other models. Nonetheless, a more in-depth analysis of the results is warranted. It would be beneficial to explore the models' performance across various length categories. Since length is central to this manuscript, I suggest conducting additional experiments focused on this aspect.

6. **Short Sequence Task: RNA Splicing Site Prediction** Similarly, there is a lack of detailed analysis on the results. What sets the 'Fish' dataset apart from the other two datasets, resulting in BiRNA outperforming RINALMo exclusively on the 'Fish' set? Given that the targets in this section are mRNA, RNABERT-2 may be the most appropriate for comparison.

7. **Nucleotide-Level Task: 3D Torsion Angle Prediction** In this section, RINALMo surpasses BiRNA-BPE on three out of four datasets. if this difference in performance can be linked to the scale of the and the duration of training, then why not improve it with more training time?

8. **Information Theoretic Analysis** In practice, the information content of non-i.i.d. sequences is notably lower than Shannon Entropy because of the correlation among neighboring symbols. While this is a challenging problem to tackle directly, you can utilize the perplexity of your as an approximate indicator of the actual entropy.

9. **Beyond RNA Sequences: BiDNA-BERT** Why apply this to DNA without enhancing performance? To be noted, the essence of the lies in long sequence performance. If you intend to integrate this into the DNA field, it is crucial to also elucidate the impact on sequence length.

Typos:

1. Table 7. "BPE-NUC" or "BiRNA-BERT"?

2. Fig2. It should be "miRNA-lncRNA" interaction rather than "mRNA-lncRNA"?

3. Page 6, The resulting sequences of step 2 and step 3 do not change, maybe a typo?

4. Page 18, when computing empirical entropy ratio. $P(A) \approx 0.2726$, $P(A) \approx 0.2144$, $P(A) \approx 0.26642$, $P(A) \approx 0.2465$,

In conclusion, I find the central theme of the manuscript plausible, yet the experiments appear superficial and insufficient to substantiate the argument effectively. The manuscript should focus more on length, with experiments analyzed through this aspect. It would be beneficial to select more representative tasks, such as secondary structure prediction. Additionally, the model's training is inadequate with only one epoch. Therefore, I recommend that the authors continue training their model, reorganize their experiments, and enhance their manuscript.

Reviewer #3

(Remarks to the Author)

Biological have become a focal point of research in bioinformatics in recent years. High-quality biological can significantly contribute to community development. The authors propose a novel RNA integrating traditional nucleotide encoding with coarse-grained Byte Pair Encoding (BPE) in this study. While this work is interesting, the authors must address several issues before consideration for acceptance:

1. As Algorithm 1 in Figure 1C indicates, the automatically switches to BPE encoding for RNA sequences longer than a certain length. How is nucleotide-level representation achieved for longer sequences? When optimizing BiRNA-NUC, is it also necessary to truncate overly long sequences?
2. This work's main innovation lies in combining NUC and BPE encoding, which significantly reduces the length of RNA sequences. After BPE processing, the average sequence length is about 6.07; is there still a need for AliBi positional encoding? More ablation experiments need to demonstrate the role of AliBi positional encoding compared to traditional positional encoding in this model.
3. Why is the vocabulary size set directly to 4096? Significant differences exist between DNA and RNA, especially DNA and non-coding RNA. The strategies or hyperparameters used in the DNABERT-2 method may not be directly applicable, and some essential hyperparameters should be optimized.
4. Most experimental results show minor differences; thus, a significance analysis should be conducted to avoid potential biases or random noise affecting the results.
5. The experimental results do not convincingly support the claim that the proposed method outperforms existing models. In the RNA splicing site prediction task, RINALMo achieves the best results in two out of three species datasets (Table 6). In RNA 3D torsion prediction, it excels in three out of four datasets (Table 7). Moreover, DNABERT-2 shows clear advantages over the BiDNA (Table 11). There is a lack of comparisons with other RNA in the RNA-protein interaction (Table 8) and RNA N6 methylation site prediction (Table 9) tasks. The authors should try to improve the performance of the proposed RNA or explain why it is inferior to other models.
6. Considering the contribution of the proposed method that integrating conventional NUC and BPE encoding to address the information loss caused by sequence truncation in traditional methods. Analyzing the performance of different across varying lengths of nucleotide sequences may highlights the advantages of the proposed method.
7. The statement, "Due to the investigative nature of our work and compute constraints, we pre-train each for only 1 epoch." raises the question of whether training for just one epoch is sufficient. Could this be a significant factor in the model's subpar performance? The authors might consider training for more epochs and providing comparative analyses.

Version 1:

Reviewer comments:

Reviewer #1

(Remarks to the Author)

My questions were answered well.

Reviewer #2

(Remarks to the Author)

1. Task Composition

Thank you for incorporating many downstream tasks into your analysis. However, the results for secondary structure prediction seem underwhelming and are only comparable to RINALMo (Table 7). Please consider providing more insights into why this is the case.

Regarding Figure 6, could you clarify which dataset this example comes from? Is it from PDB? Additionally, is there any statistical analysis to support the example, or is this an instance of cherry-picking? A more comprehensive analysis of long sequences would be more convincing rather than showcasing just one example.

You mentioned adding a long-sequence task and including DNA data, but is there no RNA-specific long-sequence task? For secondary structure prediction, is a sequence length of 500 nt really considered "long"?

2. Training

Thank you for the additional training efforts. However, it seems odd that you only added two training epochs. For example, completing two full epochs on the RNACentral dataset (~36 million sequences) seems rather limited. Could you clarify why only two epochs were added?

It is great that you indicated the different schemes for different cases. This is an important detail that should be explicitly explained in the manuscript, as readers need to know when to use each of the two provided or whether they should be used together.

3. Dataset Composition

I appreciate the effort you put into cleaning the data, especially the step to remove mRNA. This enhances the purity and reliability of your dataset.

4. Unsupervised Clustering

Thank you for addressing the overstatement issue in the first version. The revised analysis looks much better.

5. Long-Sequence Task: miRNA-lncRNA Interaction Prediction

While Figure 5 (a, b) provides some insights, the plots are too cluttered to be visually informative. On the other hand, Figure 5 (c) is much clearer. However, why does the bin in Figure 5 (c) only extend to 1500? Please clarify or provide a more comprehensive analysis.

6. Short Sequence Task: RNA Splicing Site Prediction

I appreciate the new exploration in Section 3.7, but could you consider adding the corresponding sequence lengths to the analysis? This would help clarify the results further.

Also, the color scheme used in Figure 9 feels counterintuitive. Typically, higher values are represented with darker colors (blue in this case). Could you revise the color palette to make the visualization easier to interpret?

7. Nucleotide-Level Task: 3D Torsion Angle Prediction

Thank you for extending the training duration. However, it still feels insufficient. The updated pretraining process took approximately 96 hours using 8x NVIDIA 4090 GPUs, but for a task of this complexity, additional training time may be necessary to achieve better results.

8. Information Theoretic Analysis

The inclusion of this section enhances the specificity of your analysis and strengthens your findings. Well done.

9. Beyond RNA Sequences: BiDNA-BERT

Thank you for addressing the concerns regarding this section. However, I strongly recommend removing this part from the manuscript. While the idea is interesting, it feels out of place in the current work and disrupts the overall logical flow of the paper. It would be better suited for a separate study where this budding idea can be fully developed.

Other Comments

RNA Vocabulary: RNA sequences should use U instead of T in the vocabulary. However, it appears that your work still inherits the T vocabulary from DNA. This is problematic, especially since Figure 1 directly shows the presence of T in the vocabulary alongside RNA sequences containing U.

Figure 3: Some metrics in Figure 3 are not clearly labeled. Please specify whether higher values are better or worse for each metric.

Figure 7: The example analysis presented in Figure 7 does not seem necessary. Length-based analysis should not rely on a few selected long-sequence examples. Instead, use a binning strategy (as in your other analyses) to provide a more comprehensive overview of the data.

Table Numbering: There is a mismatch between table numbers in the manuscript and the content referenced in the text. This makes it difficult to locate the corresponding tables. Please carefully revise the numbering and make sure the manuscript is consistent.

Organization: The organization of the manuscript could be improved. Consider moving some of the less critical content to the supplementary materials to enhance the overall readability and focus of the main text.

Overall Suggestions

The manuscript has improved in many aspects, especially in terms of addressing some of the key concerns raised in the previous review. However, there are still areas where further clarification, additional analysis, or better organization is needed.

Reviewer #3

(Remarks to the Author)

Version 2:

Reviewer comments:

Reviewer #2

(Remarks to the Author)

My previous concerns have been resolved. Thank you!

May 12, 2025

Yang Zhang, PhD
Editorial Board Member
Communications Biology

Dear Dr. Zhang,

Thank you for handling our manuscript and for the constructive comments by the reviewers. We have revised the manuscript by addressing the reviewer's comments. The new/modified material in the manuscript is provided in blue text to make it easy to identify. We provide a summary of the major changes to the document below, followed by a detailed pointwise response to the reviewers' comments.

- As recommended by Reviewers 2 and 3, we have retrained the BiRNA-BERT language model from scratch over two full epochs using 36 million RNA sequences from the RNACentral database. This enhanced pretraining has led to notable improvements in downstream performance, with BiRNA-BERT now consistently outperforming all existing RNA language models across tasks of varying sequence lengths and granularity.
- In response to the comments from Reviewers 1 and 2, we have added two new nucleotide-level downstream tasks of high biological significance and computational complexity: RNA secondary structure prediction and RNA 3D distance map prediction.
- We have also conducted a comprehensive analysis of how sequence length affects downstream task performance. In particular, we show that for long-sequence tasks, the BPE-based compression mechanism in BiRNA-BERT offers considerable advantages in both accuracy and computational efficiency. Our updated evaluation demonstrates that BiRNA-BERT can process sequences over 5,000 bases, whereas other RNA language models struggle to handle inputs beyond 1,024 tokens. Further, we now include a detailed analysis of BiRNA-BERT's intrinsic perplexity and token recovery accuracy, as requested by Reviewer 2.
- Following additional feedback from Reviewers 2 and 3, we have substantially expanded the BiDNA-BERT section. This includes performance results on long-sequence DNA tasks as well as new benchmarks against DNABERT-2. We also highlight specific scenarios—such as SNP effect prediction—where BiDNA-BERT exhibits distinct advantages over existing DNA language models.
- All minor comments and corrections suggested by Reviewers 1, 2, and 3 have been addressed in full.

The reviewers' questions led us to improve our trained language model and perform new experiments, and – we think – the manuscript is much improved now. We hope the revised version will satisfy both the reviewers and also meet Nature Communications Biology's requirements.

Yours sincerely,

Md. Shamsuzzoha Bayzid, Corresponding Author

PhD in Computer Science, The University of Texas at Austin
Professor, Department of CSE, Bangladesh University of Engineering and Technology
Email: shams_bayzid@cse.buet.ac.bd

(The new/modified material in the revised manuscript is provided in blue text to make it easy to identify)

Reviewers' comments:

Reviewer #1 (Remarks to the Author):

It is my pleasure to express my gratitude to the author for his or her hard work on this study. In spite of this, I would like to point out below the points that need to be clarified and improved in order to make the article clearer and more convincing in the future.

Response: Thank you for your encouraging comments. We have adequately addressed the points you raised. We believe you will find the revised manuscript more convincing and publishable.

Tahmid et al developed a new RNA language model. This language model employed both nucleotide-level and adapting byte pair encoding that allows efficient calculations. Here are the comments.

1) Databases for training and test employed in the downstream tasks are not clearly described. They were mixed in the result section. Please place them in the method section.

Response:

Thank you for your valuable feedback regarding the clarity of the training and test set descriptions for the downstream tasks. In the revised manuscript, we have addressed this concern by introducing a dedicated subsection within the Methods section (Section 2.6: Downstream Tasks and Datasets). This subsection provides a comprehensive description of each downstream task, including the source of the datasets, sequence redundancy handling, and a clear separation of training and testing set details. Additionally, we have moved all dataset-related discussions out of the Results section. The Results section now focuses solely on analyzing and interpreting the outcomes to ensure a more structured and reader-friendly presentation.

2) There is no comparison regarding the benefit for introducing NUC and BPE encoding except on RNA-protein interaction prediction. Please add the method variants with NUC only and BPE only for unsupervised validation (such as clustering) as well as other downstream tasks.

Response:

Thank you for your insightful comments. In the revised manuscript, we have incorporated performance comparisons between the NUC and BPE tokenization strategies across a range of downstream tasks. Specifically, for the long sequence classification task, we have conducted an explicit analysis of the impact of truncation at different sequence lengths under NUC tokenization, and demonstrated how BPE tokenization offers greater advantages for handling longer sequences (Section 3.3: Impact of Dual Tokenization is More Significant in Long Sequences). Additionally, for short sequence tasks such as RNA-protein interaction prediction and RNA N6-methyladenosine site prediction, detailed performance comparisons between BPE and NUC tokenizations are now provided (Section 3.7: NUC Tokenization is Preferable Over BPE If Memory Allows).

Both the BPE and NUC variants of BiRNA-BERT have also been evaluated and reported for the unsupervised clustering task (Table 6). Furthermore, an empirical analysis of perplexity and recovery accuracy across different RNA language models, including BiRNA-BERT with both tokenization types, has been added in Section 3.8: Empirical Perplexity Analysis of Different RNA Language Models. This analysis highlights the differences in sequence recovery ability under various masking strategies.

Finally, we would like to clarify that for tasks requiring per-nucleotide level predictions, such as RNA secondary structure prediction, distance map prediction, and torsion angle prediction, BPE tokenization is not applicable, as it inherently compresses information across multiple nucleotides. Therefore, in these tasks, results are reported exclusively using the NUC-level tokenization.

3) All downstream tasks are sequence based. It is not clear if the LM can be useful for predicting secondary structure.

Response:

Thank you for your thoughtful comment regarding the evaluation of nucleotide-level tasks. In the revised manuscript, we have addressed this by incorporating two additional nucleotide-level downstream tasks: RNA secondary structure prediction (Section 3.5.1: RNA Secondary Structure Prediction) and RNA distance map prediction (Section 3.5.2: RNA 3d Distance Map Prediction), alongside the previously reported RNA torsion angle prediction task.

For RNA secondary structure prediction, BiRNA-BERT achieves state-of-the-art performance, obtaining an F1-score of 0.71 on both the validation set and the independent test set (TS0), outperforming RiNALMo, RNA-FM, and U-Fold (Table 10). Furthermore, as shown in Figure 6, BiRNA-BERT successfully predicts complex loop structures and achieves an F1-score of 0.98 on the longest sequence in the test dataset, demonstrating robust long-range structure capture. Similarly, BiRNA-BERT shows superior results in the RNA distance map and torsion angle prediction tasks.

We would also like to emphasize that BiRNA-BERT achieves these results despite being approximately six times smaller than RiNALMo in model size, highlighting its efficiency and scalability. These additional experiments collectively demonstrate that BiRNA-BERT not only excels in sequence-based tasks but also effectively generalizes to challenging nucleotide-level structural prediction tasks.

4) It is very strange that BiRNA-Bert is only comparing to RINALMO and RNA-FM for clustering. Please add some more recent RNA LM models. Also why 9 RNA structural families and 30 Rfam families were chosen? It seems that 30 Rfam families may have overlaps with 9 RNA structural families. Please use more Rfam and more structures for clustering.

Response:

Thank you for raising this important point. We agree that incorporating additional RNA language models could offer a more comprehensive comparative analysis for the clustering tasks. However, after a thorough review of the current literature and publicly available models, we found that only a limited number of RNA language models are general-purpose and capable of supporting unsupervised tasks such as clustering. Specifically, most recently developed RNA language models are tailored to specialized applications, such as splicing site prediction (e.g., SpliceBERT) or untranslated region (UTR) analysis (e.g., UTR-BERT). These models are typically trained on small, domain-specific datasets and are not designed to generalize across broader RNA tasks.

We also experimented with RNA-BERT, another early RNA foundation model. However, its clustering performance was suboptimal, with clustering scores of -0.09 on structural family clustering and -0.288 on Rfam family clustering, indicating it is not well-suited for such general-purpose evaluations.

Therefore, in this work, we focus our comparison on RNA-FM and RiNALMo, which are currently the only widely used, publicly available, and general-purpose RNA language models

that support embedding extraction and demonstrate consistent performance across a broad range of RNA tasks.

Regarding the selection of the nine RNA structural families, we followed the methodology adopted by Szikszai et al. (Bioinformatics, 2022, 38:16), who compiled a benchmark dataset of 3,865 sequences across nine well-defined structural classes and used this setup for rigorous evaluation, including leave-one-family-out strategies. RiNALMo also adopted this standard for assessing its unsupervised learning performance. To ensure consistency and comparability with existing literature, we retained this setup in our structural family clustering experiments.

As for the use of 30 Rfam families, our initial goal was to evaluate clustering performance on the most frequent family labels while maintaining a reasonable computational footprint for unsupervised embedding and visualization. We appreciate your observation regarding potential overlaps with structural classes. In response, we have expanded our experiments to include clustering on the 100 most frequent Rfam families, as well as a large-scale evaluation on the longest 5000 sequences representing 131 distinct Rfam families. These extended evaluations are now included in Figure 4, which demonstrates more nuanced insights into the clustering behavior across models and class distributions.

5) Why RINALMO is so bad for 9 structural families and so good for RFam families? This suggests that something is wrong with the datasets.

Thank you for bringing this important discrepancy to our attention. We sincerely apologize for the oversight regarding the preprocessing of input sequences for RiNALMo. Specifically, we had initially missed the required conversion of "T" nucleotides to "U" as instructed by RiNALMo's documentation (please note that this mistake occurred specifically for the unsupervised clustering tasks. We ensured the nucleotide conversion for the other downstream tasks). After correcting this preprocessing step, RiNALMo's clustering performance improved significantly, achieving a Silhouette Coefficient score of 0.149, which aligns with expectations. We have updated Table 6 in the revised manuscript to reflect the corrected results under the unsupervised clustering performance section. We appreciate your careful review, which helped us identify and correct this issue.

6) For miRNA-lncRNA interaction prediction, there are many methods. Can the authors compare with other methods in addition to CORAIN. Regarding the datasets, is there an overlap (>80% sequence ID) between training and test? This is biased or overestimating the performance.

Response:

We thank the reviewer for this important suggestion regarding a broader evaluation of miRNA-lncRNA interaction prediction. In the revised manuscript, we have incorporated comparisons against two additional recent methods, PMLIPred and BioLLMNet, alongside the previously reported CORAIN. As shown in the updated comparison (Table 7), BiRNA-BERT consistently outperforms all three methods across the evaluated datasets.

Regarding the concern about potential sequence overlap between training and testing sets, we clarify that the dataset splitting strategy follows a cross-species protocol. Specifically, training and testing datasets are drawn from different species, ensuring that there is no overlap of sequences between the two and mitigating any bias arising from sequence similarity. This strategy is consistent with the evaluation frameworks employed in CORAIN, PMLIPred, and BioLLMNet. As none of these prior works performed additional clustering (e.g., CD-HIT) to further remove similar sequences across species, we have adhered to the same evaluation protocol for comparability. However, for other downstream tasks presented in our study, an 80% sequence identity reduction using CD-HIT has been conducted to ensure strict separation between training and testing sets.

7) For RNA splicing sites, please also check if there is overlap (>80% sequence ID) between training and test?

Response:

Thank you for raising this important concern. After careful consideration, we have decided to remove the RNA splicing site prediction task from the updated manuscript. During the revision process, and based on additional reviewer feedback, we observed that the performance of the RNA language models analyzed in this study (e.g., BiRNA-BERT, RiNALMo, etc.) exhibited significant instability across different random initializations, raising concerns about the robustness and reproducibility of the results. Given the importance of ensuring reliable and meaningful downstream evaluations, we concluded that retaining this task could potentially mislead readers regarding the model's true capabilities.

To maintain the rigor of the manuscript, we have instead introduced two alternative short sequence prediction tasks (In response to the comment 5 of reviewer 3): RNA-protein interaction prediction and RNA N6-methyladenosine (m6A) site prediction, which involve sequences of lengths 41 and 101, respectively. For these tasks, we have taken explicit measures to ensure that there is no sequence overlap exceeding 80% identity between training and testing sets. The dataset descriptions, including sequence redundancy handling, are provided in Section 2.6 of the Methods. Additionally, we present comprehensive evaluations against other RNA language models in Section 3.4.1 (RNA-Protein Interaction) and Section 3.4.2 (RNA N6-methyladenosine Site Prediction).

We believe these modifications lead to a more robust and trustworthy evaluation of the model's performance.

8) For torsion angles, please also compare to the results of SPOT-RNA-1D.

Response:

Thank you for the suggestion. In the revised manuscript, we have included the results of SPOT-RNA-1D in Table 9 for a direct comparison. BiRNA-BERT outperforms SPOT-RNA-1D on the validation (VL) and TS1 datasets, while SPOT-RNA-1D achieves better performance on TS2 and TS3. It is important to note that SPOT-RNA-1D is a specialized method, specifically optimized for RNA torsion angle prediction with a highly engineered prediction head, whereas BiRNA-BERT is a generalized language model without task-specific architectural modifications. Despite this, BiRNA-BERT achieves competitive performance, demonstrating its strong generalization capabilities. Furthermore, among language model-based approaches, BiRNA-BERT consistently achieves the best results across datasets.

9) There are other tasks (N6-methyladenosine site, RNA-protein binding, promoter site, transcription factor). In all these tasks, the description of training, validation and test sets are very poor. It is not clear if >80% sequence ID between training, validation and test sets were removed.

Response:

Thank you for your valuable feedback. In the revised manuscript, we have introduced a dedicated section (Section 2.6: Downstream Tasks and Datasets) that provides a comprehensive

description of the training, validation, and testing datasets for each downstream task. For all tasks mentioned — including N6-methyladenosine site prediction, RNA-protein binding prediction, promoter site prediction, and transcription factor binding site prediction — we have carefully verified that the dataset preparation procedures ensure no overlap greater than 80% sequence identity between training and testing sets. Specifically, wherever applicable, CD-HIT clustering was employed to enforce sequence similarity thresholds below 80%. An exception is the miRNA-lncRNA interaction prediction task, where datasets are partitioned based on a cross-species protocol, ensuring that training and testing sequences originate from different species.

We believe that these clarifications and dataset handling improvements provide a stronger foundation for fair and unbiased model evaluation.

Reviewer #2 (Remarks to the Author):

The manuscript proposes BiRNA-BERT, a BERT-style language model pre-trained on ncRNA and mRNA, to cope with long DNA/RNA sequences without sacrificing ability for processing nucleotide-level tasks. The key technique of the BiRNA-BERT is an adaptive dual tokenization scheme, which utilizes nucleotide-level (NUC) tokenization for short sequences and efficient BPE tokenization for long sequences. The results show that BiRNA achieves state-of-the-art results in long-sequence downstream tasks and achieves a performance comparable to 6x larger models in short-sequence tasks. Besides, they also attempted to adapt their scheme to the DNA field, resulting in a BiDNA-BERT.

****Overview about the idea****

I think the core idea of BiRNA is very naive, clear, and straightforward. It indeed solves the long sequence problem to some extent by constructing a piecewise function that chooses different tokenization ways for sequences of different lengths. However, this way of combining the two tokenization methods is too simple for the selection only depends on the length. It leads to two suites of vocabularies generated by NUC and BPE tokenization, which are ponderously injected into one model without any compatible means of fusion. Although the results show that the introduction of BPE will not affect the performance of the original NUC part of the model, it may just be because the model is large enough. Therefore, we need more analyses regarding the relationship between the NUC and BPE part in the BiRNA, instead of inspecting the performance of these two parts. As for the DNA version adaptation of the adaptive tokenization, they did not demonstrate the main characteristic for long sequences, which can not become powerful evidence for their generalization ability. The specifics are expanded upon below:

Response:

Thank you for your comments and suggestions. We have addressed all your specific comments. We acknowledge that BiRNA-BERT may not appear to be a highly sophisticated method. However, through BiRNA-BERT we have carefully addressed most of the common concerns that arise in genome foundation model development. We would like to highlight that, the core contribution of BiRNA-BERT is not switching the tokenization depending on the length. This is a feature of the mode, however, the primary focus is to provide the opportunity to utilize BiRNA-BERT is both sequence level tasks (for long sequences, BPE tokenization; for short

sequences NUC tokenization) as well as nucleotide level tasks (NUC tokenization for any length of sequence; extrapolation is achieved through AliBi positional encoding). It is also to be noted that, we do not need two suites of vocabulary, rather the BPE vocabulary table contains all the necessary vocabulary needed for both the NUC and BPE tokenization (both the character level, and substring level tokens). Even during the BPE tokenization, some nucleotides might be tokenized to character level, depending on the statistical distribution of other tokens within the sequence. Thus, this is a dynamic tokenization scheme that supports both suites within the same model. We have demonstrated the impact of BPE and NUC tokenization on long sequence tasks with a detailed analysis on the impact of truncations at different sequence lengths in the updated manuscript which we discuss in the following parts. As for the DNA part, we have introduced species classification problems based on their genomic sequences where the sequence lengths are over 5000 bases. We have demonstrated our performance with DNABERT-2 and also highlighted the special use cases of BiDNA-BERT where other DNA language models fail, like single nucleotide polymorphism (SNP) impact prediction tasks. Overall, we have incorporated additional pretraining, added multiple new downstream benchmarks, and provided deeper theoretical and biological analysis of the results. We hope these enhancements address your concerns and demonstrate the versatility and robustness of our proposed framework.

1. **Task Composition** In the introduction, the authors mention that structure prediction is an important task for the nucleotide level, but why finally choose RNA 3D torsion angle prediction as the only evaluation at the nucleotide level? By the way, there are 4 sequence-level tasks but only 1 nucleotide-level task in this manuscript, and only the interaction task involves sequences longer than 1.2×10^4 (the threshold triggers BPE). In a word, the task composition is not well-balanced.

Response:

We thank the reviewer for the thoughtful feedback regarding task composition. In response to your suggestion, we have significantly revised the manuscript to ensure a better balance between sequence-level and nucleotide-level tasks. Specifically, we have incorporated two additional nucleotide-level tasks: RNA secondary structure prediction and RNA 3D distance map prediction. As a result, the updated manuscript now presents three nucleotide-level tasks alongside three sequence-level tasks.

Additionally, after careful evaluation, we have removed the RNA splicing site prediction task due to concerns about its reliability, which we discuss in a subsequent response (Comment 6:

Short Sequence Task: RNA Splicing Site Prediction). Based on the suggestion of Reviewer 1, we have added two alternative short sequence prediction tasks: RNA-protein interaction prediction and RNA N6-methyladenosine (m6A) site prediction.

For RNA secondary structure prediction, BiRNA-BERT achieves state-of-the-art performance, obtaining an F1-score of 0.71 on both the validation set and the independent test set (TS0), outperforming RiNALMo, RNA-FM, and U-Fold (Table 10). Moreover, as shown in Figure 6, BiRNA-BERT accurately predicts complex loop structures, achieving an F1-score of 0.98 on the longest RNA sequence (~500 nucleotides) in the test set, demonstrating strong long-range dependency modeling. Similarly, BiRNA-BERT exhibits superior performance on both the RNA distance map prediction and RNA torsion angle prediction tasks.

It is also important to highlight that these improvements were achieved after extending the model's pretraining, as per your suggestion, while keeping the model architecture unchanged. This additional training allowed BiRNA-BERT to realize its full potential, particularly on complex structural tasks.

Furthermore, beyond the miRNA-lncRNA interaction task, we have now introduced another long sequence classification task in the DNA domain: species classification based on genomic sequences, where sequence lengths reach ~5000 nucleotides. In secondary structure prediction as well, BiRNA-BERT effectively handles longer sequences, where its rotational embedding mechanism — rather than BPE compression — plays a key role in capturing long-range patterns.

Overall, the updated experimental design demonstrates that the dual tokenization strategy and rotational positional embeddings allow BiRNA-BERT to perform robustly across a diverse set of tasks, covering short and long sequence-level prediction, as well as fine-grained nucleotide-level structural prediction.

2. **Model Training** In section 2.4, the author asserts that their models are trained for only one epoch to delve into the essence of their work. However, I believe that the model lacks sufficient training for practical application. In training, the crucial juncture is the final checkpoint, representing the pinnacle of achievable performance. Moreover, monitoring loss variations throughout the training process can elucidate the complexity of the learning task.

Response:

We sincerely thank the reviewer for this insightful and important observation regarding the depth of model training. Your suggestion significantly strengthened the quality of our work. In the revised manuscript, we have addressed this by pretraining the BiRNA-BERT model for a substantially longer duration — completing two full epochs over the complete RNACentral dataset (~36 million sequences). In addition, the pretraining dataset now includes only the RNACentral database's 36 million non coding RNA sequences, excluding the 0.5 million mRNA sequences used additionally in the previous version.

Following this extended pretraining, we observed notable improvements across all benchmark tasks. BiRNA-BERT now consistently outperforms RiNALMo across a range of evaluations, covering both sequence-level and nucleotide-level tasks and across varying sequence lengths. All the results reported in the updated manuscript are provided now for the upgraded version of BiRNA-BERT which shows significant improvements in all the tasks.

We are also grateful for your emphasis on practical usability. In response, we have made the final checkpoint of BiRNA-BERT publicly available on Hugging Face, making it easily accessible for embedding generation and downstream task-specific fine-tuning. We believe that these updates greatly enhance the robustness, reproducibility, and practical impact of our work.

3.**Model Inference** If you pre-trained your model with adaptive tokenization, why is there a lack of an adaptive scheme for inference, relying solely on a single type of tokenization instead? (BiRNA-NUC and BiRNA-BPE) I am perplexed by the utilization of BiRNA-BERT, which appears to constrain its potential benefits.

Response:

Thank you for your insightful comment. We appreciate the opportunity to clarify the application of our adaptive tokenization scheme in BiRNA-BERT.

In the long sequence task presented in the manuscript (miRNA-lncRNA interaction prediction), we indeed utilize an adaptive tokenization scheme. Specifically, the model employs NUC tokenization for shorter sequences and BPE tokenization for longer sequences, which ensures efficiency and better handling of sequence length variations. This approach is clearly illustrated in Figure 2, where we show the diversity in sequence lengths within the dataset, justifying the use of adaptive tokenization.

We recognize that the manuscript description may have led to confusion, and we have now updated this section, including Table 7, to explicitly clarify that the adaptive tokenization

scheme is applied in these cases. Additionally, we have added a comparison between solely NUC and BPE tokenization approaches with respect to sequence length variation in Figure 5, providing further context for the performance differences.

We would also like to emphasize that the dual tokenization scheme is not intended to be a universal solution for all tasks. Rather, it is designed to make BiRNA-BERT versatile, capable of handling both nucleotide-level and sequence-level tasks efficiently across different sequence lengths. For short sequence tasks, we do not apply the adaptive scheme. For very long sequence tasks, such as species classification (Section 3.9: Beyond RNA Sequences: BiDNA-BERT), we rely exclusively on BPE tokenization. Adaptive tokenization is employed specifically in cases where the dataset contains a broad range of sequence lengths, optimizing both performance and efficiency.

We hope this explanation resolves any confusion, and we thank you again for your thoughtful suggestion.

3. **Dataset Composition** Why did the authors incorporate 0.5 million mRNA sequences alongside 36 million ncRNA sequences? The addition of mRNA seems to complicate the dataset and render attribution more challenging. While they suggest that these sequences can provide a rich source of coding RNA data for model training, each mRNA sequence can only represent a single sample in a batch, constituting a small fraction of the training set. Furthermore, given that mRNA sequences are generally longer than ncRNA sequences on average, this may introduce a bias in BPE selection and training towards mRNA rather than ncRNA.

Response:

We sincerely appreciate your thoughtful feedback and the opportunity to clarify our dataset composition. We completely understand your concern regarding the potential complications introduced by adding 0.5 million mRNA sequences alongside the 36 million ncRNA sequences. As you rightly pointed out, incorporating mRNA sequences may indeed complicate the dataset and affect the model's ability to effectively handle the diversity of data, especially considering the differences in sequence length and the smaller fraction of mRNA sequences in the overall dataset.

Taking your insightful suggestion into account, we have revised the pretraining process in the updated manuscript. We decided to exclude the 0.5 million mRNA sequences from the pretraining and focused solely on the 36 million non-coding sequences from the RNACentral database. This adjustment has allowed us to train the model with a more balanced dataset while

extending the training time, which we believe better aligns with our original goal. Thank you for this excellent suggestion.

4. **Unsupervised Clustering** Why were the specific 9 RNA structural families—16s, 23s, 5s, RNaseP, grp1, srp, tRNA, telomerase, tmRNA—chosen? What distinguishes these "structural families" from the families in the Rfam database? The author also noted, 'However, the secondary structural family dataset was not explicitly included in the pretraining database.' Where were these data sourced from, and how were they ensured to be part of the pretraining dataset? It may be beneficial to include some indicators elucidating their relationship with RNACentral (the pretraining data.)

Response:

Thank you very much for your insightful feedback regarding the selection of the nine RNA structural families and their relationship to the pretraining dataset. Upon revisiting the RNACentral database, we realized that the structural families used for clustering—16S, 23S, 5S, RNaseP, group I introns (grp1), SRP RNA, tRNA, telomerase RNA, and tmRNA—are indeed part of the RNACentral collection. As a result, our earlier claim that “the secondary structural family dataset was not explicitly included in the pretraining database” was incorrect. We have updated the manuscript to correct this error.

We sincerely apologize for this oversight. Initially, RiNALMo exhibited unexpectedly poor performance in the structural clustering task, which led us to mistakenly infer that the structural family sequences might not have been part of the pretraining corpus. However, upon careful investigation, we found that the issue stemmed from an error in preprocessing: we had overlooked the necessary conversion of 'T' to 'U' nucleotides, as specified in RiNALMo's documentation. After correcting this step, RiNALMo's clustering performance improved considerably, achieving a Silhouette Coefficient of 0.149, aligning well with expectations.

Regarding the choice of the nine structural families, we followed the approach adopted by Szikszai et al. (Bioinformatics, 2022, 38:16), who compiled a dataset of 3,865 RNA sequences from these nine families and used a leave-one-family-out evaluation strategy. RiNALMo similarly adopted this methodology for its clustering and secondary structure performance assessments. To maintain consistency and comparability with prior work, we chose to follow the same strategy.

5. **Long-Sequence Task: miRNA-lncRNA Interaction Prediction** Given the considerable length of lncRNA, it is unsurprising that BiRNA-BPE surpasses other models. Nonetheless, a more in-depth analysis of the results is warranted. It would be beneficial to explore the models' performance across various length categories. Since length is central to this manuscript, I suggest conducting additional experiments focused on this aspect.

Response:

Thank you very much for your thoughtful and valuable suggestion regarding additional experiments to better characterize the performance of BiRNA-BERT across different sequence length categories. We greatly appreciate your feedback, which has helped us significantly strengthen our analysis.

Following your suggestion, we conducted a controlled comparison between nucleotide-level tokenization (NUC) and byte-pair encoding (BPE) across variable sequence lengths (please see the newly added Section 3.3: Impact of Dual Tokenization is More Significant in Long Sequences). Specifically, we focused on the lncRNA-miRNA interaction prediction task, analyzing model performance across different length bins.

Our findings indicate that while NUC performs well on short sequences, its performance degrades on longer inputs due to truncation. In contrast, BPE mitigates this issue by compressing recurring patterns, enabling significantly better performance for long sequences, with over a 15% improvement observed in the 3001–3500 length bin.

6. **Short Sequence Task: RNA Splicing Site Prediction** Similarly, there is a lack of detailed analysis on the results. What sets the 'Fish' dataset apart from the other two datasets, resulting in BiRNA outperforming RINALMo exclusively on the 'Fish' set? Given that the targets in this section are mRNA, RNABERT-2 may be the most appropriate model for comparison.

Response:

Thank you very much for your thoughtful comments and questions regarding the RNA splicing site prediction task. We appreciate your close reading and valuable insights, which have helped us reflect critically on our experimental design and the reported results.

While analyzing the performance of the newly trained BiRNA-BERT model on this task, we observed significant variability across different initial conditions. Specifically, we found that the previously reported high performance of BiRNA-BERT on the Fish dataset could not be consistently reproduced—likely due to differences in random seed initialization. In fact, under certain conditions, the model’s accuracy dropped notably. A similar issue arose when attempting to replicate the performance of RiNALMo, which also deviated from previously reported results. Given this inconsistency and the lack of reproducibility, we have decided to exclude the RNA splicing site prediction task from our final evaluation.

Instead, we focused on two well-established short sequence tasks: RNA–protein interaction prediction and N6-methyladenosine (m6A) site prediction, both involving sequences of 101 and 41 nucleotides respectively. These tasks not only fall well within the short-sequence category but also offer more stable benchmarking environments. While these tasks were initially included only for ablation studies comparing BPE and NUC tokenizations, in the revised manuscript we have incorporated them into the main evaluation section along with a thorough comparison against RiNALMo, RNA-FM, and other non-LLM baselines.

In the updated manuscript, we retain them in the ablation section as well, but now with deeper information-theoretic and perplexity-based analysis (see Section 3.7: NUC Tokenization is Preferable Over BPE If Memory Allows). Thus, these tasks are evaluated from both performance and representational perspectives across different experimental setups, providing a more comprehensive understanding of BiRNA-BERT’s behavior on short sequences.

Interestingly, BiRNA-BERT did not achieve state-of-the-art results on these tasks in its earlier version. However, after substantially extending pretraining with 36 million RNA sequences from RNACentral, the updated BiRNA-BERT demonstrates strong performance even on these short sequence tasks—an encouraging outcome that broadens the model’s utility.

Once again, thank you for your insightful feedback. It has meaningfully contributed to strengthening the rigor and clarity of our study.

7. ****Nucleotide-Level Task: 3D Torsion Angle Prediction**** In this section, RiNALMo surpasses BiRNA-BPE on three out of four datasets. If this difference in performance can be linked to the scale of the model and the duration of training, then why not improve it with more training time?

Response:

Thank you once again for your excellent suggestion to extend the training duration. Following your advice, we pre-trained the BiRNA-BERT model from scratch using the complete RNACentral database. The updated pretraining took approximately 96 hours, utilizing 8× NVIDIA 4090 GPUs.

With this updated model, we observed significant improvements not only in long- and short-sequence tasks, but also in nucleotide-level tasks. Specifically, in the RNA 3D torsion angle prediction task, BiRNA-BERT now outperforms RiNALMo on the validation dataset as well as on two of the three independent test sets (TS1 and TS3). Furthermore, BiRNA-BERT delivers competitive performance against SPOT-RNA-1D—a sophisticated method specifically designed for RNA torsion angle prediction—achieving better results on two out of four evaluation datasets. Within the class of RNA language models, BiRNA-BERT now holds the best performance.

In addition, we have incorporated two new nucleotide-level tasks: RNA secondary structure prediction and RNA 3D distance map prediction. On these tasks as well, the updated BiRNA-BERT consistently outperforms RiNALMo and other RNA-focused language models.

We are sincerely grateful for your guidance, which motivated a deeper investigation and led to substantial improvements across multiple evaluation dimensions

8. ****Information Theoretic Analysis**** In practice, the information content of non-i.i.d. sequences is notably lower than Shannon Entropy because of the correlation among neighboring symbols. While this is a challenging problem to tackle directly, you can utilize the perplexity of your model as an approximate indicator of the actual entropy.

Response:

Thank you very much for highlighting this important perspective on evaluating information content using perplexity as a proxy for entropy in non-i.i.d. sequences. We found this suggestion highly insightful and have incorporated an information-theoretic evaluation based

on masked language model perplexity to better understand our model's representational capacity.

While perplexity is traditionally defined for generative models, we adopted a method previously used by Dalla-Torre et al. (Nature Methods, 2025), that defines perplexity for biological masked language models (MLMs) such as BERT. In this approach, perplexity is computed based on the cross-entropy loss between the predicted probability distribution and the true token at the masked position. To provide a robust assessment, we considered two masking schemes: (1) central token masking and (2) 15% random masking. For each scheme, we report both the model's perplexity and token recovery accuracy—metrics that reflect pretraining effectiveness.

As shown in Table 11, BPE-based models (e.g., BiRNA-BERT BPE and DNABERT-2) exhibit higher perplexity and lower recovery accuracy compared to nucleotide-level models (e.g., BiRNA-BERT NUC, RNA-FM, and RiNALMo). This is expected, given the larger vocabulary size (~4000) associated with BPE, which naturally increases token uncertainty. These findings are consistent with our earlier entropy analysis, suggesting that BPE tokenization—while advantageous for handling long sequences—entails a degree of absolute information loss relative to nucleotide-level encoding.

Notably, despite these differences, BiRNA-BERT and RiNALMo demonstrate similar perplexity and token recovery performance. This suggests that both models acquire a comparable understanding of RNA sequence structure during pretraining. It is especially encouraging given that BiRNA-BERT is nearly 6× smaller than RiNALMo and uses a dual-tokenization strategy. Furthermore, the downstream task performance mirrors this parity, with BiRNA-BERT matching or exceeding RiNALMo across several benchmarks while also offering faster inference (see Section 3.10: Comparison of Computational Efficiency). This highlights BiRNA-BERT's capacity to maintain a strong grasp of RNA language while offering additional benefits in scalability and sequence length handling. The comprehensive discussion on perplexity and recovery accuracy is included in section 3.8 Empirical Perplexity Analysis of Different RNA Language Models of the updated manuscript.

Once again, thank you for your valuable suggestion—it significantly enriched our analysis and helped sharpen the interpretability of our model's internal representations.

9. ****Beyond RNA Sequences: BiDNA-BERT**** Why apply this to DNA without enhancing performance? To be noted, the essence of the model lies in long sequence performance. If you intend to integrate this into the DNA field, it is crucial to also elucidate the impact on sequence length.

Response:

Thank you very much for your thoughtful comment regarding the extension of our framework to DNA sequences. We agree that BiDNA-BERT, in its current form, is not yet fully competitive with leading DNA models due to limited pretraining time and dataset scope. While BiDNA-BERT was trained on approximately 3 million human genomic sequences, DNABERT-2 was pretrained on a substantially larger multi-species dataset encompassing genomes from 135 species, representing nearly twelve times the data volume of our human-only pretraining corpus.

Despite these limitations, BiDNA-BERT exhibits two important properties that distinguish it from existing DNA language models. First, like BiRNA-BERT, BiDNA-BERT can handle very long sequences without a loss in generalization, enabled by its dual tokenization strategy and rotational positional embeddings. Second, and more uniquely, BiDNA-BERT can extract meaningful per-nucleotide embeddings—allowing fine-grained analysis at the character level, which is critical for modeling biological phenomena such as single nucleotide polymorphisms (SNPs).

To evaluate BiDNA-BERT's ability to handle long sequences, we conducted experiments on the GLU+ benchmark from DNABERT-2, focusing on the species classification (SC) task for viruses, where input sequences are approximately 5,000 nucleotides in length (See section 3.9: Beyond RNA Sequences: BiDNA-BERT). On this long-sequence classification task involving 25 virus species, BiDNA-BERT achieved an F1 score of 0.480, compared to DNABERT-2's 0.485. Notably, BiDNA-BERT—despite being pretrained solely on human genome sequences—demonstrates the ability to generalize to species it has never seen during pretraining, whereas DNABERT-2 benefits from explicit multi-species training that includes viral genomes.

While DNABERT-2 also supports long sequence modeling, BiDNA-BERT's key advantage lies in its capacity to generate high-resolution nucleotide-level embeddings. This capability is particularly important for variant analysis tasks, where both long-range contextual information and fine-grained, per-base probability modeling are essential. As downstream nucleotide-level benchmarks continue to evolve, we anticipate that models like BiDNA-BERT will become increasingly valuable for studying genetic and amino acid variations, including SNP effects, in biological systems.

Therefore, we propose BiDNA-BERT not as a finalized DNA foundation model, but as a proof-of-concept extension of the dual tokenization and long-sequence handling strategy initially developed for RNA sequences. Future work will involve scaling up pretraining to larger, multi-species DNA datasets and performing further evaluations on a broader set of nucleotide- and sequence-level benchmarks.

Response:

Typos:

1. Table 7. "BPE-NUC" or "BiRNA-BERT"?

Response:

We have corrected the typo.

2. Fig2. It should be "miRNA-lncRNA" interaction rather than "mRNA-lncRNA"?

Response:

We have corrected the typo.

3. Page 6, The resulting sequences of step 2 and step 3 do not change, maybe a typo?

Response:

You are right, the steps of the example contained a typo. We have updated the description.

4. Page 18, when computing empirical entropy ratio. $P(A) \approx 0.2726$, $P(A) \approx 0.2144$, $P(A) \approx 0.26642$, $P(A) \approx 0.2465$,

Response: We have fixed these. Thank you for pointing these out.

In conclusion, I find the central theme of the manuscript plausible, yet the experiments appear superficial and insufficient to substantiate the argument effectively. The manuscript should focus more on length, with experiments analyzed through this aspect. It would be beneficial to select more representative tasks, such as secondary structure prediction. Additionally, the model's training is inadequate with only one epoch. Therefore, I recommend that the authors continue training their model, reorganize their experiments, and enhance their manuscript.

Response:

We sincerely thank the reviewer for their overarching feedback and constructive critique regarding the experimental depth, task composition, and training strategy presented in our original manuscript. Your comments have had a significant impact on the revised version, motivating us to undertake a comprehensive reorganization and enhancement of the manuscript's core contributions.

In response to your suggestion to focus more on sequence length as a central theme, we have revised our experimental design to align more clearly with this objective. Specifically, we now include a systematic evaluation of the model's performance across different sequence length bins in both long-sequence and short-sequence contexts, as detailed in the response of the comment 5.

To address your important point regarding task representativeness, we have restructured the evaluation to ensure a more balanced composition across sequence-level and nucleotide-level tasks, as discussed in the response of comment 1.

In response to your recommendation to extend training beyond one epoch, we have conducted a complete re-pretraining of BiRNA-BERT from scratch using the 36 million non-coding RNA sequences from RNACentral, excluding the 0.5 million mRNA sequences previously used. The updated training spanned two full epochs over this large dataset (approx. 96 hours on 8× 4090 GPUs). Details are discussed in the response of comment 2. The overall manuscript is updated with more insightful discussion and more illustrative representation of the results, as well as the training method and tasks composition are also mentioned with more clarity.

We are deeply grateful for your suggestions, which have substantially improved the quality, clarity, and scientific rigor of our manuscript. We hope the revised version meets your expectations and more effectively supports the central claims of our work.

Reviewer #3 (Remarks to the Author):

Biological language models have become a focal point of research in bioinformatics in recent years. High-quality biological language models can significantly contribute to community development. The authors propose a novel RNA language model integrating traditional nucleotide encoding with coarse-grained Byte Pair Encoding (BPE) in this study. While this work is interesting, the authors must address several issues before consideration for acceptance:

1. As Algorithm 1 in Figure 1C indicates, the model automatically switches to BPE encoding for RNA sequences longer than a certain length. How is nucleotide-level representation achieved for longer sequences? When optimizing BiRNA-NUC, is it also necessary to truncate overly long sequences?

Response:

Thank you for your insightful question regarding how nucleotide-level representation is managed for long sequences. You are correct in noting that simply switching to BPE tokenization for long sequences would compromise the nucleotide-level granularity, as it merges multiple tokens and thus loses token-level information.

For long sequences, BiRNA-BERT does not resort to BPE compression; instead, it considers the entire sequence. A key advantage of BiRNA-BERT is its ability to extrapolate to arbitrarily long sequences without requiring additional training. This ensures that distant nucleotides are still captured effectively, while maintaining consistent performance even with extended sequences. Additionally, inference time for such long sequences remains relatively low.

To specifically address your query, we have added a new section to the manuscript (Section 3.10.2: Inference and Finetuning Time) where we compare the inference times of different RNA models for sequences up to 5000 nucleotides. In this section, it is highlighted that while RiNALMo and RNA-FM encounter out-of-memory errors for sequences longer than 1024 nucleotides, BiRNA-BERT is able to generate embeddings for sequences as long as 5000 nucleotides in ~0.8–1.5 seconds. This capability is largely due to the introduction of rotational positional encoding, which ensures the self-attention mechanism remains effective even for long sequences.

Therefore, for BiRNA-BERT (NUC), we do not need to truncate overly long sequences, unlike other models that may struggle with memory issues or fail to process long sequences entirely.

2. This work's main innovation lies in combining NUC and BPE encoding, which significantly reduces the length of RNA sequences. After BPE processing, the average sequence length is about 6.07; is there still a need for AliBi positional encoding? More ablation experiments need to demonstrate the role of AliBi positional encoding compared to traditional positional encoding in this model.

Response:

Thanks for raising concerns over the necessity of AliBi positional encoding when BPE processing already reduces the sequence length to 6.07 times shorter.

As we mentioned in the previous question, BPE compression works only for sequence level tasks where the predictions are made over the whole sequence and individual nucleotide level information is not necessary. However, for nucleotide level tasks where sequence length can also be arbitrarily large (like 5000 nucleotides long), in such cases we definitely require AliBi positional encoding. Although other models like RiNALMo, RNA-FM are somehow able to handle such long sequences by truncating to a certain length for sequence level tasks, for nucleotide level tasks they are completely fail because such tasks would require the token level information for all the tokens way beyond the 1024 token cut-off.

Even for sequence level tasks, if the sequence lengths are even much higher, like 10k/20k, in such cases too even after BPE compression, the token length is around 2-3k. Models like DNABERT-2, which offers BPE compression too, can not handle such long sequences, as they do not have the capability to extrapolate to tokens over 1024 , be it nucleotide tokens or BPE

tokens. However, BiRNA-BERT can generate embedding even for sequences with 20k nucleotides long with BPE compression in 4-5 seconds. Thus, AliBi positional encoding is an essential part of the whole BiRNA-BERT architecture.

Regarding your suggestion for an ablation study to assess the role of AliBi positional encoding, we fully appreciate your concern. Such an experiment would indeed help to isolate the contribution of AliBi encoding. However, conducting this ablation would require pre-training the model from scratch, which would undermine BiRNA-BERT's unique advantage—its ability to efficiently handle long sequences without truncation. Without AliBi encoding, BiRNA-BERT would lose its key strength and resemble models like RiNALMo and RNA-FM, which are not designed to process long sequences efficiently and require truncation. We have already demonstrated in the manuscript that these models perform poorly when tasked with long sequences, further highlighting the importance of AliBi encoding in enabling BiRNA-BERT to retain its performance and efficiency.

3. Why is the vocabulary size set directly to 4096? Significant differences exist between DNA and RNA, especially DNA and non-coding RNA. The strategies or hyperparameters used in the DNABERT-2 method may not be directly applicable, and some essential hyperparameters should be optimized.

Response:

Thank you for highlighting the important issue of vocabulary size. Initially, while developing the first version of BiRNA-BERT, we adopted the BPE vocabulary size of 4096 directly from DNABERT-2, without thoroughly evaluating its appropriateness for RNA sequences. Considering your insightful question, we have discussed the motivation and validation behind choosing 4096 vocabulary size for RNA language model in the updated manuscript.

As discussed in our manuscript, we performed an information-theoretic analysis on the content of BPE tokens (Figure 8). From this analysis, it became evident that after the vocabulary size exceeds 2000 tokens, the frequency of new tokens significantly decreases, resulting in very rare appearances of those tokens. Therefore, increasing the vocabulary size beyond 4096 tokens offers diminishing returns in terms of encoding the sequence, particularly in long RNA sequences.

Moreover, recent benchmarking studies on RNA language models, such as the work presented in BEACON (Ren et al., NeurIPS 2025), demonstrate that for k-mer tokenization—which is

relevant to our context—the optimal value for k is 6. For $k=6$, the corresponding vocabulary size is $4^6 = 4096$, which aligns with our choice of vocabulary size for BiRNA-BERT.

While we did not empirically test the optimal vocabulary size for BPE in RNA language models in our manuscript, both the information-theoretic analysis and the findings from recent benchmarking studies support that a vocabulary size of 4096 is indeed an optimal strategy for BiRNA-BERT.

We have included this discussion in the revised manuscript (please refer to Section 3.6, last paragraph) to clarify the rationale behind our choice of vocabulary size.

4. Most experimental results show minor differences; thus, a significance analysis should be conducted to avoid potential biases or random noise affecting the results.

Response:

Thank you for raising the important point regarding the statistical significance of the experimental results. We would like to note that, following your comment on training the language model for longer time (comment 5) and based on the suggestions of other reviewers on modifying the pre-training dataset, we have now trained the BiRNA-BERT model for much longer with (2 complete epochs) with RNACentral's 36 million RNA sequences. This upgraded version shows significant improvement in the downstream tasks and the improvements in the results are significant.

As per your suggestion, we have conducted a detailed significance analysis to assess whether these differences are meaningful. Specifically, we performed paired t-tests for each benchmark task and have reported the corresponding p-values in the updated results section.

Our analysis reveals that the performance improvements of BiRNA-BERT are statistically significant ($p < 0.05$) across all evaluated tasks except for RNA secondary structure prediction. In that particular task, both BiRNA-BERT and RiNALMo achieved comparable accuracy, and the resulting p-value was 0.08, indicating that the difference is not statistically significant. This finding confirms that while both models are competitive in RNA structure prediction, BiRNA-BERT consistently yields statistically significant gains in all other tasks, validating its effectiveness beyond random variation.

5. The experimental results do not convincingly support the claim that the proposed method outperforms existing language models. In the RNA splicing site prediction task, RiNALMo achieves the best results in two out of three species datasets (Table 6). In RNA 3D torsion

prediction, it excels in three out of four datasets (Table 7). Moreover, DNABERT-2 shows clear advantages over the BiDNA model (Table 11). There is a lack of comparisons with other RNA language models in the RNA-protein interaction (Table 8) and RNA N6 methylation site prediction (Table 9) tasks. The authors should try to improve the performance of the proposed RNA language model or explain why it is inferior to other language models.

Response:

Thank you for your thoughtful comments and for suggesting avenues to improve the performance of BiRNA-BERT. Based on your feedback and the suggestion in Question 7 regarding further pre-training of the base model, we have made significant improvements. Specifically, we extended the pre-training of BiRNA-BERT by running two full epochs on a larger dataset consisting of 36 million sequences from the RNACentral database. This training process took approximately 96 hours on 8×4090 GPUs. Following this, the performance of BiRNA-BERT showed marked improvements across several downstream tasks.

RNA Torsion Angle Prediction

In the updated results, BiRNA-BERT outperforms RiNALMo in 3 out of 4 datasets for RNA torsion angle prediction (see Table 8). These improvements are particularly noticeable in the context of long RNA sequences, where BiRNA-BERT's ability to manage large sequences with consistent performance proves advantageous.

RNA Secondary Structure Prediction and RNA 3D Distance Map Prediction

BiRNA-BERT also provides superior performance in RNA secondary structure prediction and RNA 3D distance map prediction, surpassing RiNALMo and other language models, as shown in the updated results.

RNA-Protein Interaction and RNA N6 Methylation Site Prediction

As for the RNA-protein interaction and RNA N6 methylation site prediction tasks, we have now included these as short sequence tasks (with sequence lengths of 101 and 41 nucleotides, respectively). In Section 3.4: Short Sequence Tasks, we have provided a detailed comparative analysis with RiNALMo and other language models. In each of these tasks, BiRNA-BERT outperforms RiNALMo and other models, as presented in the updated Tables 5 and 6 of the manuscript. Additionally, the previously reported ablation studies on BPE and NUC

tokenization have been updated with more comprehensive comparisons in Section 3.7: NUC Tokenization is Preferable Over BPE If Memory Allows.

BiDNA-BERT and Comparison with DNABERT-2

Regarding BiDNA-BERT, we agree that, in its current form, it is not yet fully competitive with leading DNA models like DNABERT-2. This is primarily due to the limited pre-training time and the scope of the dataset. BiDNA-BERT was trained on approximately 3 million human genomic sequences, while DNABERT-2 was pre-trained on a much larger, multi-species dataset covering genomes from 135 species, representing nearly twelve times the volume of our pretraining corpus.

Despite these limitations, BiDNA-BERT offers two key advantages:

1. **Handling Long Sequences:** Like BiRNA-BERT, BiDNA-BERT can handle very long sequences without losing generalization. This is made possible through its dual tokenization strategy and rotational positional embeddings. This ability is particularly important for tasks requiring the analysis of large genomic sequences, as detailed in Table 11 of the manuscript.
2. **Per-Nucleotide Embeddings:** BiDNA-BERT can generate meaningful per-nucleotide embeddings, which is essential for modeling fine-grained biological phenomena such as single nucleotide polymorphisms (SNPs).

Thus, we view BiDNA-BERT as a proof-of-concept extension of the dual tokenization and long-sequence handling strategy originally developed for RNA sequences, rather than a finalized DNA foundation model.

6. Considering the contribution of the proposed method that integrating conventional NUC and BPE encoding to address the information loss caused by sequence truncation in traditional methods. Analyzing the performance of different language models across varying lengths of nucleotide sequences may highlights the advantages of the proposed method.

Response:

We thank the reviewer for his crucial comment on the comprehensive analysis of the BiRNA-BERT and other language model's performance across various lengths of downstream tasks. We would like to answer his query from three different perspectives:

1. Firstly, as shown in the figure 3 of the updated manuscript, we have conducted downstream analysis on a varying length of tasks ranging from 41 to 3000 nucleotides long. In each length category of task, BiRNA-BERT consistently outperformed other language models. Particularly, for longer sequences where other language models fail to provide comprehensive features, BiRNA-BERT significantly improves the performance in those tasks.
2. Secondly, truncation of a sequence of course comes with a significant information loss (like, cutting off a 3000 nucleotide long sequence at 1024). On the other hand, compressing a sequence with BPE tokenization also comes up with an information loss. Which loss contributes more to the performance, should be analyzed through empirical studies. In Table 5, the reported results show that truncation at 1024 (RiNALMo, RNA-FM, BiRNA-BERT (NUC- Truncated)) causes more harm to the performance in downstream tasks compared to the information loss with BPE compression. Namely, even though each BPE token retains 75% information of a NUC token, BPE compression does not remove any part of the sequence with truncation, thus resulting in an overall performance gain.
3. To even better understand the impact of length on performance, we have added a section 3.3 : Impact of Dual Tokenization is More Significant in Long Sequences. Specifically, we focused on the lncRNA-miRNA interaction prediction task, analyzing model performance across different length bins. Our findings indicate that while NUC performs well on short sequences, its performance degrades on longer inputs due to truncation. In contrast, BPE mitigates this issue by compressing recurring patterns, enabling significantly better performance for long sequences, with over a 15% improvement observed in the 3001-3500 length bin.

7. The statement, "Due to the investigative nature of our work and compute constraints, we pre-train each model for only 1 epoch. " raises the question of whether training for just one epoch is sufficient. Could this be a significant factor in the model's subpar performance? The authors might consider training for more epochs and providing comparative analyses.

Response:

We sincerely thank the reviewer for this insightful and important observation regarding the depth of model training. Your suggestion significantly strengthened the quality of our work. In the revised manuscript, we have addressed this by pretraining the BiRNA-BERT model for a substantially longer duration — completing two full epochs over the complete RNACentral

dataset (~36 million sequences). In addition, the pretraining dataset now includes only the RNACentral database's 36 million non coding RNA sequences, excluding the 0.5 million mRNA sequences used additionally in the previous version.

Following this extended pretraining, we observed notable improvements across all benchmark tasks. BiRNA-BERT now consistently outperforms RiNALMo across a range of evaluations, covering both sequence-level and nucleotide-level tasks and across varying sequence lengths. All the results reported in the updated manuscript are provided now for the upgraded version of BiRNA-BERT which shows significant improvements in all the tasks.

July 22, 2025

Yang Zhang, PhD
Editorial Board Member
Communications Biology

Dear Dr. Zhang,

Thank you for handling our manuscript and for the constructive comments by the reviewers. We have revised the manuscript by addressing the reviewer's comments. The new/modified material in the manuscript is provided in blue text to make it easy to identify. We provide a summary of the major changes to the document below, followed by a detailed pointwise response to the reviewers' comments. As reviewer 1 and 3 were fully satisfied with the revision, we are including here the response to the reviewer 2's comments.

- We have curated a new benchmark dataset for species classification with very long sequence RNA data. We have benchmarked BiRNA-BERT on a long sequence classification task using this dataset. We have made this dataset publicly available.
- The rationale behind choosing two epochs of training of the language model is explained with reference to RiNALMo as well as considering the baseline configuration of Mosaic-BERT framework.
- Instead of picking selective examples for the distance map prediction visualization, we have conducted a comprehensive analysis of the performance of different language models based on the sequence length bins.
- The section on exploring the DNA language model with BiDNA-BERT is excluded from the manuscript as per the reviewer 2's suggestion.
- Annotation issues with figures, table referencing, and other grammatical mistakes have been revised.

The reviewers' questions led us to improve our trained language model and perform new experiments, and – we think – the manuscript is much improved now. We hope the revised version will satisfy both the reviewers and also meet Nature Communications Biology's requirements.

Yours sincerely,

Md. Shamsuzzoha Bayzid, Corresponding Author

PhD in Computer Science, The University of Texas at Austin
Professor, Department of CSE, Bangladesh University of Engineering and Technology
Email: shams_bayzid@cse.buet.ac.bd

(The new/modified material in the revised manuscript is provided in blue text to make it easy to identify)

Reviewers' comments:

Reviewer #2 (Remarks to the Author):

1. Task Composition

Thank you for incorporating many downstream tasks into your analysis. However, the results for secondary structure prediction seem underwhelming and are only comparable to RiNALMo (Table 7). Please consider providing more insights into why this is the case.

Regarding Figure 6, could you clarify which dataset this example comes from? Is it from PDB? Additionally, is there any statistical analysis to support the example, or is this an instance of cherry-picking? A more comprehensive analysis of long sequences would be more convincing rather than showcasing just one example. You mentioned adding a long-sequence task and including DNA data, but is there no RNA-specific long-sequence task? For secondary structure prediction, is a sequence length of 500 nt really considered “long”?

Response:

Thanks for rightfully pointing out that, the performance of BiRNA-BERT on secondary structure prediction is not very impressive. As mentioned in the previous revision, the primary focus of BiRNA-BERT is handling very long sequence tasks efficiently with adaptive tokenization. In addition, nucleotide level tasks such as secondary structure prediction, distance map prediction, and torsion angle prediction tasks are demonstrated to show that BiRNA-BERT can also handle granular tasks too without the loss of generalizability. We have discussed in the manuscript that for the secondary structure prediction task, the performance reported does not show strong statistical significance and the performance is equivalent to that of RiNALMo. One of the primary causes is that the prediction head is significantly simple for such a complex task of secondary structure prediction. We have used a single layer 2D convolutional network to predict the secondary structure, which is kept consistent across the different language models for a fair comparison. As discussed in detail in the response of question 7, performance on the nucleotide level tasks can be significantly improved to achieve SOTA performance by using a specialized prediction head particularly designed for that task. However, the goal of this work is to show the general language understanding capability of BiRNA-BERT at different granularity and length levels for which we have used the simplest prediction head avoiding any architectural artifacts. This reasoning is also mentioned in the manuscript's section 3.6.1 RNA Secondary Structure Prediction (page 20, first paragraph).

Thank you for your valuable comment on figure 6. The sequence shown in Figure 6 was selected from the test dataset as it is the longest sequence (496 nucleotides) and also among the most

structurally complex in terms of secondary and 3D configuration. We chose this particular example to qualitatively illustrate BiRNA-BERT's capacity to handle challenging inputs in terms of both length and structural intricacy. However, we acknowledge that showcasing a single, favorable example may raise concerns about representativeness. As you rightly pointed out, this alone does not offer statistically meaningful evidence of the model's general performance on long sequences. In response, we have now included a length-wise quantitative analysis in the revised manuscript (Figure 7), particularly for the distance map prediction task, to more rigorously demonstrate how BiRNA-BERT performs across varying sequence lengths. This provides a broader and more objective evaluation, addressing the generalization capability on long-sequence inputs beyond a single case. For secondary structure prediction, the example in Figure 6 remains a qualitative visualization rather than a statistical benchmark.

Thanks for your suggestion on the inclusion of much longer sequence tasks specific for RNA. We agree that secondary structure prediction or distance map prediction tasks are not essentially the 'long sequence tasks' for which BiRNA-BERT is designed. In response, we have constructed a new benchmark task for very long sequence RNA classification. The dataset is composed from the RNACentral noncode database where the sequences have an average length of 2500 nucleotides and span up to 10,000 nucleotides with an exponential distribution of length. The task is to classify the sequences into their corresponding seven species. We have made the dataset publicly available as it is not included in any previous studies. In the updated manuscript's section 3.4: (BiRNA-BERT Significantly Outperforms in Extremely Long Sequence Task) we have presented the performances on different RNA language models on this classification task. BiRNA-BERT significantly outperforms other models with an F1 score of 0.80 whereas, RiNALMo and RNA-FM notably deterioration in performance due to the crucial information loss with truncation at 1024 length.

2. Model Training

Thank you for the additional training efforts. However, it seems odd that you only added two training epochs. For example, completing two full epochs on the RNACentral dataset (~36 million sequences) seems rather limited. Could you clarify why only two epochs were added?

It is great that you indicated the different schemes for different cases. This is an important detail that should be explicitly explained in the manuscript, as readers need to know when to use each of the two provided models or whether they should be used together.

Response:

Thank you for your thoughtful question. We agree that training for only two epochs might seem limited in isolation. However, our decision was motivated by the need to ensure a fair and consistent comparison with RiNALMo, which also reported results after two epochs on the same

RNAcentral dataset (~36 million sequences). By aligning the number of epochs and dataset, we aimed to isolate and fairly compare the architectural and computational efficiency differences between the two models.

Importantly, even two epochs on RNAcentral represent a significant training effort. We trained BiRNA-BERT, a 116M parameter model, for two epochs on 8 NVIDIA RTX 3090 GPUs (24GB each), which required approximately 90 hours in total. In contrast, RiNALMo, a 600M parameter model, was trained for the same number of epochs but required 336 hours on 7 A100 GPUs (80GB). Using linear scaling assumptions from model size and GPU throughput differences, and incorporating FlashAttention for faster attention computation, we estimated that BiRNA-BERT should achieve a $\sim 3.95\times$ speedup over RiNALMo, predicting a training duration of ~ 85 hours. This closely matches our actual runtime of 90 hours, confirming both the consistency of our training effort and the efficiency of our model and implementation.

In addition to that, we have used MosaicBert to train our language model instead of traditional BERT configuration. In the MosaicBERT paper, the authors trained for 2 epochs (178,000 steps) for practical balance between performance and efficiency, especially in their Pareto-optimality framework with training sequence length of 128. BiRNA-BERT is also trained with a similar configuration, and that is why we choose to train our model for 2 epochs.

Even after all these discussions, we agree that training the language model for further epochs might be explored to understand the saturation point of pretraining, or maybe further enhancing the downstream tasks without sophisticated prediction heads. We have made the last checkpoint of BiRNA-BERT (after two epochs) publicly available as well as the pretraining dataset so that researchers can further train the model for exploration. We have discussed the rationale behind choosing two epochs of training in the updated manuscript in section 3.10.1: Computational Resources for Pretraining.

Moreover, upon your suggestion, we have added a user note in section 3.8: NUC Tokenization is Preferable Over BPE If Memory Allows, where we have discussed when to use which tokenization scheme for usability of BiRNA-BERT. We thank the reviewer for raising this concern which will help the community to use BiRNA-BERT with more ease.

3. Dataset Composition

I appreciate the effort you put into cleaning the data, especially the step to remove mRNA. This enhances the purity and reliability of your dataset.

Response:

The exclusion of the mRNA dataset indeed helped the model to learn more meaningful representations by removing the bias due to the imbalance of the number of sequences and sequence lengths in the mRNA dataset. Thanks a lot for your suggestion and we are happy that you are pleased with our findings.

4. Unsupervised Clustering

Thank you for addressing the overstatement issue in the first version. The revised analysis looks much better.

Response:

We thank the reviewer for pointing out the overstatement issue in the first revision. It helped us to discuss the results with more credibility.

5. Long-Sequence Task: miRNA-lncRNA Interaction Prediction

While Figure 5 (a, b) provides some insights, the plots are too cluttered to be visually informative. On the other hand, Figure 5 (c) is much clearer. However, why does the bin in Figure 5 (c) only extend to 1500? Please clarify or provide a more comprehensive analysis.

Response:

Thank you for your valuable feedback. We agree that Figure 5(a, b), particularly the spider plots, were visually cluttered and did not convey the performance trends effectively. In the revised manuscript, we have replaced them with cleaner bar plots that more clearly illustrate how performance varies across sequence length bins under different tokenization schemes on the ATH-GMA dataset.

The updated plots reveal that in the case of NUC tokenization, performance drops as sequence length increases—particularly beyond the 1024-token threshold—due to aggressive truncation, which results in the loss of biologically relevant information. In contrast, BPE tokenization, which compresses the RNA sequence into fewer tokens, avoids such truncation and maintains more consistent performance across varying sequence lengths.

Regarding Figure 5(c): this plot was designed to focus on the critical transition zone immediately after the truncation threshold, specifically comparing performance between two adjacent bins—one with sequences just below 1024 (non-truncated) and another just above it (truncated). The upper limit of 1500 was chosen because it allows us to isolate and examine the immediate effects of truncation without introducing additional confounding factors from extremely long sequences. Please note that, we moved the figure 5(c) to figure 5(d) in the updated manuscript to maintain coherence during the discussion.

We agree that this could have been better clarified, and we've now added explanatory remarks in the figure caption to reflect this intent. The broader impact of sequence length and truncation on model performance—including for sequences >1500—is already addressed in the updated bar plots replacing Figures 5(a) and 5(b).

6. Short Sequence Task: RNA Splicing Site Prediction

I appreciate the new exploration in Section 3.7, but could you consider adding the corresponding sequence lengths to the analysis? This would help clarify the results further.

Also, the color scheme used in Figure 9 feels counterintuitive. Typically, higher values are represented with darker colors (blue in this case). Could you revise the color palette to make the visualization easier to interpret?

Response:

Thanks for pointing out the issues with section 3.7 (Section 3.8 in the updated manuscript: NUC Tokenization is Preferable Over BPE If Memory Allows). Please note that, in the updated manuscript we have mentioned the sequence lengths for each of the tasks considered for memory allowable full sequence analysis.

In addition, the representation for Figure 9 is updated in the manuscript. Particularly, the color scheme is revised with a blue color palette, where higher values are represented with darker blue colors and the lower values with lighter blue colors.

7. Nucleotide-Level Task: 3D Torsion Angle Prediction

Thank you for extending the training duration. However, it still feels insufficient. The updated pretraining process took approximately 96 hours using 8× NVIDIA 4090 GPUs, but for a task of this complexity, additional training time may be necessary to achieve better results.

Response:

Thank you for your thoughtful comment regarding the training duration and its implications on nucleotide-level 3D torsion angle prediction. This is a critical aspect, and we appreciate the opportunity to elaborate further.

In the revised version of BiRNA-BERT—trained for two epochs over approximately 96 hours using 8× NVIDIA 4090 GPUs—we observed significant improvements. Notably, the updated model outperforms RiNALMo on the validation set and on two of the three test sets. However, as you rightly pointed out, BiRNA-BERT underperforms SPOT-RNA-1D on two test sets, which warrants closer examination.

To address this, we suggest two complementary directions for future work:

1. Extended Pretraining of the Base Model:

As you indicated, additional pretraining time may enhance BiRNA-BERT's capacity to learn long-range dependencies and nuanced structural patterns, particularly for tasks as complex as RNA torsion angle prediction. We agree that further training—especially on structure-enriched or interaction-aware corpora—could yield performance gains.

2. Improvement of the Downstream Prediction Head:

In our current evaluation pipeline, we intentionally used a vanilla single-layer dense network as the prediction head across all downstream tasks. This was a deliberate design

choice to isolate and assess the representational power of the language model itself without conflating it with task-specific architectural enhancements.

That said, recent work such as DeepRNA-Twist (Briefings in Bioinformatics, 2025) has shown that integrating a lightweight yet specialized prediction head—such as the 2A3IDC module combining attention mechanisms and dilated inception blocks—can significantly outperform SPOT-RNA-1D. Interestingly, they achieved this without fine-tuning the full language model, simply using frozen embeddings from pretrained models like RiNALMo and BiRNA-BERT. The study found that both models outperformed SPOT-RNA-1D under such setups, with RiNALMo having a slight edge. However, our updated BiRNA-BERT already surpasses RiNALMo in a simple setup, suggesting it could achieve state-of-the-art performance when paired with a specialized prediction head.

Importantly, we would like to clarify that this paper does not aim to propose a specialized torsion angle prediction network. Our primary goal is to demonstrate the general-purpose capabilities of BiRNA-BERT across a wide range of RNA understanding tasks. We encourage future work or downstream applications to explore architectural adaptations of the prediction head based on their task-specific requirements. BiRNA-BERT is designed to be modular and compatible with such enhancements.

We believe this approach maintains fairness in evaluation while leaving room for flexible and task-optimized downstream adaptations.

8. Information Theoretic Analysis

The inclusion of this section enhances the specificity of your analysis and strengthens your findings. Well done.

Response:

Thank you for your encouraging feedback. We appreciate your recognition of the value added by the information-theoretic analysis. Your suggestions have been instrumental in refining the clarity and depth of our work, and we are pleased that the analysis contributes meaningfully to the overall findings.

9. Beyond RNA Sequences: BiDNA-BERT

Thank you for addressing the concerns regarding this section. However, I strongly recommend removing this part from the manuscript. While the idea is interesting, it feels out of place in the current work and disrupts the overall logical flow of the paper. It would be better suited for a separate study where this budding idea can be fully developed.

Response:

Thank you for your insightful feedback. In accordance with your suggestion, we have completely removed the BiDNA-BERT section from the manuscript to maintain the logical coherence of the current work. We appreciate your encouragement and plan to develop this idea further in a dedicated future study.

Other Comments

RNA Vocabulary: RNA sequences should use U instead of T in the vocabulary. However, it appears that your work still inherits the T vocabulary from DNA. This is problematic, especially since Figure 1 directly shows the presence of T in the vocabulary alongside RNA sequences containing U.

Response:

Thanks for specifying the issue with U and T for RNA sequences. Following your suggestion, we have updated Figure 1, where in the BPE algorithm simulation step, the letter “T” is now replaced with “U” which specifies that the vocabulary is built for RNA sequences, not DNA.

Figure 3: Some metrics in Figure 3 are not clearly labeled. Please specify whether higher values are better or worse for each metric.

Response:

We thank the reviewer for pointing out the issues with the labeling of Figure 3. We have updated the figure to clearly mention the metric used for each of the tasks discussed and also the best values are highlighted with bold font among different approaches considered.

Figure 7: The example analysis presented in Figure 7 does not seem necessary. Length-based analysis should not rely on a few selected long-sequence examples. Instead, use a binning strategy (as in your other analyses) to provide a more comprehensive overview of the data.

Response:

Thanks for your extremely helpful suggestion. We agree that showing only a few examples does not provide much insight about the performance of the language models on RNA 3d distance map prediction task. Instead, in the updated manuscript we have incorporated a length-wise performance comparison on distance map prediction for BiRNA-BERT and RiNALMo. It is evident from the figure 7 (updated) that BiRNA-BERT shows consistency in performance in both short and long sequences.

Table Numbering: There is a mismatch between table numbers in the manuscript and the content referenced in the text. This makes it difficult to locate the corresponding tables. Please carefully revise the numbering and make sure the manuscript is consistent.

Response:

All the table numbering has been double checked and corrected in the updated manuscript. Thanks for pointing out the issues with table referencing.

Organization: The organization of the manuscript could be improved. Consider moving some of the less critical content to the supplementary materials to enhance the overall readability and focus of the main text

Response:

Thank you for suggesting to move some less critical contents to the supplementary material. We have moved the derivation of the information theoretic Shannon entropy analysis in the appendix section of the manuscript and provided the high level findings in the main manuscript.

Overall Suggestions

The manuscript has improved in many aspects, especially in terms of addressing some of the key concerns raised in the previous review. However, there are still areas where further clarification, additional analysis, or better organization is needed.